# Correlation Analysis between Roadway Networks and Economic Ranking—Case Study: Municipalities and Departments of Colombia

**Carlos Felipe Urazán-Bonells [1], Maria Alejandra Caicedo-Londoño [1,\*] and Hugo Alexander Rondón-Quintana [2]** 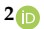

1    Programa de Ingeniería Civil, Facultad de Ingeniería, Universidad de La Salle, Bogota 111711, Colombia
2    Facultad del Medio Ambiente y Recursos Naturales, Universidad Distrital Francisco José de Caldas, Bogota 111711, Colombia
*    Correspondence: macaicedo@unisalle.edu.co; Tel.: +57-3178896874

**Abstract:** It is generally assumed that there is a statistically valid correlation between the length of a roadway network, in addition to other factors such as its classification and/or average travel speed, and economic indicators such as Gross Domestic Product (GDP) and the Municipal Relative Weight (MRW), considering that the roadway network and transport development generate economic development in a region. This study reports the results of correlating several variables which are economic indicators of roadway networks, both at a municipal and a departmental level, in Colombia; it was concluded that at the level of municipalities, there is no valid correlation between MRW, as a dependent variable, and the average travel speed and the sum of the length (in kilometers) of the roadways that connect villages, as independent variables. There was a correlation with neither the MRW as an independent variable nor the traveling distance and time for each municipality concerning the capital city of each respective department. Finally, it was found that the department agribusiness GDP was associated with the length of the tertiary roadway network and with the primary network, with an R2 of 0.7. This study concludes that activities in rural zones are the ones that generate the greatest impact on roadway investment within a region.

**Keywords:** Gross Domestic Product; tertiary roadway network; Municipal Relative Weight

## 1. Introduction

Colombia is a country that has an urgent need to revitalize rural activity, making it more productive. The adequate transportation of agricultural products is essential for this development, and this is why it is necessary to deepen the study of the variables by which economic investment in improving rural roads is decided.

When studying the relationship between the economic development of a country's regions and the condition and quality of the road network, it is found that several studies show the existence of a relationship between roadway networks and the economic condition of a region or territory, especially in rural areas. However, few studies present a relationship between the condition of a roadway network and a macroeconomic variable such as Gross Domestic Product (GDP). Urazan et al. (2017) [1] concluded that for a sample of countries that are representative of Latin America and the Caribbean, there was a statistically valid correlation between the length of a primary-level roadway network of the analyzed countries with their GDP. Contrary to the above, the independent variables, the second-level roadway network, and the paved network did not statistically correlate with GDP.

Jacoby (2000) [2] described the important role of roadways in rural development. In the case of Nepal, he concludes that providing extensive roadway connections to agricultural markets leads to important benefits to poor homes located in the area. However, he clarifies

that these benefits are not enough to be able to conclude the achievement of reducing poverty or inequality.

Escobal and Ponce (2002) [3], mentioned that the relationship between poverty reduction and the availability of rural infrastructure has broadly been discussed since the decade of 1990. According to them, studies conducted in South East Asia demonstrate the existence of a relationship between massive investment in rural roadways and posterior economic growth, as well as a reduction in rural poverty. Jalan y Ravallion (2002) [4] concluded that economic progress in rural areas not only depends on the development of roadway infrastructure, but it also depends on other basic infrastructures such as electric power.

Meerzman and Nazemzadeh (2017) [5] established that in the case of Europe (Belgium), the development of transport infrastructure as an economical-improvement driver is not always constant. Important differences may be present between the regions of one country, especially given the presence or absence of other elements which favor this economic growth.

Howe and Richards (2019) [6] reported that in the South Sahara area there is no clarity in the relationship between investment in rural roadways (associated with lower transport costs) and greater agricultural productivity, which implies better economic conditions for the community. This paper also studies cases in India and concludes that improvements in agricultural production occurred through technological improvements, and do not have a strong relationship with roadway improvements for accessibility. In the case of Japan, the authors mentioned, that there have been important improvements in agricultural production with few roadway kilometers to travel, and that economic growth in rural areas is not necessarily caused by agricultural activities. In other words, there are other types of activities that contribute to economic activity, and therefore, to GDP.

Asher and Novosad (2020) [7] concluded that, at a worldwide level, close to one billion people live in rural areas with no access to a national roadway network connected through paved roads. In the case of India, several years after the construction of rural-access roadways, their main effect has been facilitating the arrival of agricultural workers; however, this did not cause an improvement in economical income or greater productivity. Furthermore, even with better roadway connections to markets, remote rural areas may continue to lack better economic opportunities.

Nautiyal and Sharma (2021) [8] proposed several criteria for conducting roadway maintenance. The main and most critical criterion is the improvement of the asphaltic layer, which affects average travel speed. An assessment of the pavement condition was followed by an analysis of the area of influence's social and economic activity, as well as its connectivity to other communities. According to the authors, the rural roadway network has a significant effect on the development of a nation.

In an analysis of economic growth in Colombia, Cortés and De la Peña (2019) [9] stated that the increase in transport infrastructure is essential to increase competitiveness within countries that are members of the Pacific Alliance. They determine Colombia to be lagging in its load-transport infrastructure in roadways, railways, and ports; this is not surprising given that investment in this sector has been 3.2%, when the recommendation of multinational entities is to invest close to 6%. Regarding the quality of infrastructure, Colombia is behind Peru, Chile, and Mexico.

Palei (2015) [10], based his arguments on World Bank statistics, states that one of the main factors that relates to the economic development and competitiveness of a country is the quality of its roadways, railway infrastructure, and airport infrastructure, as well as the electric power supply network.

Bottasso et al. (2021) [11], studied the case of Brazil, and established that the length of a roadway infrastructure between the years 2005–2015 significantly helped to generate greater investment in load-transport systems, especially in remote areas and maritime ports.

Idei and Kato (2018) [12] researched the effects of improvements in roadways on rural markets. The authors mentioned in their state-of-the-art review that there is no established relationship between the state of roadways and economic benefit in homes within the

roadway's influence area. In the case of Cambodia, they concluded that good condition in roadways implies a greater number of commutes regarding agricultural and livestock products, mainly those transported by bicycle or motorcycle.

Using multivariable models, Gosh and Dinda (2021) [13] analyzed the relationship between transport infrastructure and economic growth in India (analysis period 1990–2017). The study relates the indicators of three transport systems with the per capita GDP in India. In the main conclusion, the authors mention that air and roadway transport have positive long-term impacts and that railway transport is not important.

In the case of China, Magazzino and Mele (2020) [14] studied 28 regions with data from 1990 to 2017, concluding that transport infrastructure investment differs from region to region and affects economic growth to the aggregate level.

Working in Pakistan, Batool and Goldmann (2020) [15] studied the relationship between private and public investment in transport infrastructure to promote more private investment, which results in better values of GDP.

Rokicki and Stepniak (2018) [16] concluded that the investment in transport infrastructure in Poland between 2004 and 2014 resulted in a better accessibility indicator but was not statistically significant with the growth of regional productivity in urban areas, and negatively correlated with growth in rural areas.

Studying the Wider Economic Impacts (WEI), Rothengatter (2017) [17] concluded that the assessment of investment in transport infrastructure on GDP is only valid in industrialized countries. In another context, Hansen and Johansen (2017) [18] studied the economic impacts of several transport infrastructure projects in Norway, also using the wider economic impacts, but setting as travel variables the commuting and migration flows between regions.

In the case of Colombia, a document of the Consejo Nacional de Política Económica y Social (National Council for Economic and Social Policy) (CONPES, 2016) [19] shows, in Graph 4, that there has been a trend of greater government investment in rural roadways within territories with a greater rural population. However, Graph 3 shows that there was no average investment trend in territories where there is a greater length of the tertiary roadway network. Escobar et al. (2016) [20] mentioned that it is widely acknowledged that in any region of Colombia, having a good roadway network is related to good economic levels and increased quality of life for the inhabitants. However, the article does not analyze the correlation between GDP and the Roadway Network Density Index (RNDI).

In summary, there are not many studies that analyze the correlation between the quantity and condition of roadway networks of a municipality or department (states or autonomous communities in the case of other countries) with variables that measure the economic level within a territory. Several of the cases found that analyze the relationship between transport infrastructure and economic growth do not differentiate between urban and rural areas. A greater portion of studies supports the idea of a contribution of rural roadway networks to rural development or territorial progress with the function of improving roadway conditions. A large number of studies support the idea that improving transit conditions on rural roads, contributes to rural development, that is, to territorial progress. This study contributes to the state of the art, by attempting to correlate travel conditions (time and average speed) and roadway length with economic development indicators such as departmental Gross Domestic Product (GDP) and Municipal Relative Weight (MRW) for municipalities.

The purpose of the study is to perform a basic statistical analysis of the correlations between data, as a first approach to establishing the economic variables to be studied in greater detail, and to understand that explain the global impact of transport infrastructure in rural areas. This will contribute in an important way to the state of knowledge in Latin America. There are not many studies that analyze the correlation between the quantity and conditions of the roadway networks of a municipality or department with variables that measure the economic level within a territory. Aiming at this point, this paper contributes to the state of the art, by attempting to correlate travel conditions (time, average speed)

and roadway length with economic development indicators such as departmental Gross Domestic Product and Municipal Relative Weight for municipalities.

The hypothesis put forward in this manuscript is that the extent of a territory's rural network is correlated, on large scale, with its economic productivity.

## 2. Materials and Methods

### 2.1. Hypothesis and Object of Study

This study proposes a hypothesis that states that the good condition of the roadway network of in a territory must correlate with good economic conditions within it. Likewise, the opposite effect of such a correlation would be expected if the roadway network was not in good condition. A study published by Urazan et al. (2017) [1] was the foundation that motivated us to propose this hypothesis. Given the above, it was thought that this relationship could occur if applied on a national level (Colombian case), correlating information at municipal and departmental levels. This study analyzed the following:

A.    At a municipal scale, a sample of 247 cases, distributed across 5 departments, with records of roadway length through the shortest route between municipalities and each of their rural villages and districts, and the corresponding travel speed (taken in the field with the use of Global Positioning System (GPS) equipment, during working day hours).

B.    The same sample at a municipal scale, but with travel times, distance, and average travel speed. Similar analysis to the previous one, but on On this occasion, samples the data were taken between each municipality and the capital city of that department. These records were georeferenced during Labor Day hours.

C.    A sample Analysis at the departmental level, adding data from 247 municipalities and grouping them into each of the corresponding 5 departments.

D.    A sample Analysis at the departmental level, but this time for 27 departments in the country, using the following information: length of the paved roadway network and length of the non-paved roadway network. The length was analyzed separately according to the following conditions: Very Good, Good, Regular, Poor, and Very Poor. Work also included the length of tertiary and primary roadway networks.

The following are variables that indicate economic condition: the MRW of each municipality in its department, the global GDP of each department, and the sector GDP for agribusiness and mining activities. Several regressions were carried out, combining all the independent variables (length of paved and non-paved roadway network) to evaluate whether there was a statistically valid correlation (R2 coarse adjustment and/or above 0.7) (Dagnino, 2014) [21], (Pando and San Martín, 2004) [22], (Roy et al. 2020) [23], to define whether any of the characteristics of the roadway network explain or correlate with the economic condition of a municipality within its respective department or a department at a national level.

### 2.2. Methodology

This study was initially applied to municipalities, rural villages, and districts that belong to 5 departments in Colombia, of which 4 (Cundinamarca, Meta, Quindio, and Casanare) conducted travel to obtain average speed values, distance, and travel times, using GPS tracking apps. The records obtained were compared to travel times appearing in Google Maps, and we concluded that there is an average difference of only 5%. This value allows us to attest that the information provided by Google Maps is reliable. In the fifth department's case (Santander), the average travel time and distance were recorded, using only information from Google Maps. Fieldwork and its analysis were conducted as the graduation project of a group of Civil Engineering students from Universidad De la Salle (Bogota, Colombia). The results of these projects are described in García and Silva (2017) [24]—Cundinamarca and Casanare; Carrascal and Cuervo, (2019) [25]—Meta; Marín and Fonseca (2021) [26]—Quindío; Insuasti and Pérez (2021) [27]—Santander (Tables A1–A5 in Appendix A).

The variables (obtained on the field) that were initially analyzed for each department were the following: average travel speed and distance between the municipality village or district. As such, with the average speed of all the studied municipalities, an average value was established for each department. The total kilometer distance was also calculated for each municipality to its rural villages and districts, and the total kilometer distance of each municipality was added to obtain an average value per department.

It is important to highlight that the study was only carried out for roadways that connect to the 247 municipalities selected (in the previously cited graduation projects) with their respective rural villages and districts. Given the difficulties of cost and time for carrying out travel, in the Casanare department, work was conducted with 18 municipalities, in Cundinamarca with 113 municipalities, and Santander (using Google Maps) with 81 municipalities, including their respective departments' capital cities.

As such, the variable that defines the economic position of municipalities within a department is the Municipal Relative Weight in the departmental aggregate value (%) (MRW) (Tables A1–A5). The data were obtained from the National Administrative Statistics Department—*Departamento Administrativo Nacional de Estadística DANE* Departamento Administrativo Nacional de Estadística DANE (2020) [28]. The mentioned indicator was selected given that there are no GDP records per municipality.

The information was analyzed using a parametric statistical study, with correlations performed to check, initially, whether the $R^2$ value was close to or greater than 0.7 to validate the relationships between the data. In that case, the analysis was continued by checking the standard error and *p*-value results to complete the parametric study. If the analysis was with multiple independent variables ($\times 1$, $\times 2 \ldots \times n$), the adjusted $R^2$ value was verified. The analysis was conducted using Microsoft Excel software. Software such as "R" or similar could also be used.

Based on the study's hypothesis, the lengths of the roadways that connect municipalities to their villages and districts was correlated with the average travel speed obtained by tracking those routes as independent variables, with the MRW as a dependent variable. For each correlation we verified whether the adjusted $R^2$ was equal to or greater than 0.7. If the above occurred, we verified the coefficient's sign for each of the independent variables to validate its condition of being direct or inverse concerning the dependent variable. If this condition was not met with the adjusted $R^2$, the correlation was discarded. This analysis was carried out to analyze whether a correlation of the economic condition of municipalities occurred with the lengths of their rural roadway networks and their circling conditions (average travel speed; greater speed was interpreted as better roadway condition).

It is important to highlight that there is a statistical correlation between one variable and another if when one of them changes value, the other changes too. A way to check whether there is an important correlation between variables is the $R^2$ value because is a measure of how far the plot points fall from the regression line. In other words, it is calculated with the vertical distance from each point to the trend line. If these distances are small and consistent, the $R^2$ value is large and close to 1. The contrary is true if the value is small and close to 0. A value of $R^2$ equal to or greater than 0.7 means that there is a strong relationship between the correlated data [29–31]. Remember that the $R^2$ value increases or decreases the level of confidence about the relationships among the hypothetical variables.

If an acceptable $R^2$ is obtained, the validity of the correlation is verified with an analysis of the standard error and *p*-value. The present study only checked that the *p*-value results were equal to or less than 0.05, and a low value for the standard error, in case the $R^2$, was equal to or greater than 0.7. Similar criteria were applied to the R value. If the $R^2$ value result was statistically validated, we proceeded to confirm this using R coefficient analysis.

The statistical analysis and determination of correlations between data in this study were only the first approach to establishing the economic variables to be studied in greater detail to understand the global impact of transport infrastructure in rural areas.

A new analysis was conducted in which the arithmetic average of travel time and distance between each municipality was added; however, in this case, the department's capital

city was an independent variable that explained the MRW (Tables A1–A5 in Appendix A). This new analysis is proposed in case there is a good correlation between the economic conditions of municipalities and their roadway connections to the capital city of their department, given that the capital is the main social and economic connection node.

Then, an aggregate analysis was carried out for municipalities in each of the 5 departments. Correlations between the departmental GDP, average travel speed, and kilometer distance values for village connection roadways were analyzed for each department (Table A6).

As such, we decided to broaden the sample to 27 departments (out of a total of 32 in Colombia) for which there is information available related to roadway conditions, through the Instituto Nacional de Vías de Colombia (INVIAS) [32]. For this purpose, the analysis used departmental GDP information as a dependent variable (DANE, 2020) [28], and it used roadway network length—according to its condition (paved or non-paved roadways) with a rating of Very Good, Good, Regular, Poor, or Very Poor—as an independent variable (Table A7). The data used corresponded to those of 27 INVIAS territories, which correspond to the following departments: Antioquia, Atlántico, Bolívar, Boyacá, Caldas, Caquetá, Casanare, Cauca, Cesar, Chocó, Córdoba, Cundinamarca, Guajira, Huila, Magdalena, Meta, Nariño, Norte de Santander, Putumayo, Quindío, Risaralda, Santander, Sucre, Tolima, Valle, and San Andrés.

Lastly, a similar analysis to the above was carried out; however, it specified the contribution of the GDPs of greatest relevance to the economic sectors in a rural scenario: agriculture, livestock, fishing, hunting, and silviculture (presented together as the agribusiness sector), and additionally, added the GDP value of mining (Table A8). Additionally, other correlations were carried out between the agribusiness with mining GDP and the length of the primary- and tertiary-level roadway networks (INVIAS, 2020) [28], for each department (Table A9).

## 3. Results

### 3.1. Analysis According to Municipality

The first regressions carried out were for the municipalities selected in each department, using MRW as a dependent variable, and average travel speed and the sum of the kilometer distances for village connection roadways as independent variables. For the department of Casanare, the adjusted $R^2$ for multiple linear regression was (0.062) for Cundinamarca (−0.009), for Meta (−0.064), for Santander (0.36), and for Quindío (−0.039). Low R2 values led us to discard the correlation of MRW with these two independent variables.

Given that the analysis above was carried out made with both independent variables and was not statistically valid did not result in statistically feasible outcomes, the correlation was only carried out with the lengths of roadway networks that connect villages and districts, proposing the hypothesis that if there exists a greater number of kilometers of connections to villages and districts of a given municipality, there is a greater distance to travel, negatively impacting in rural productivity and, therefore, being related to a lower MRW value. The correlations between roadway network length and MRW were carried out for the following regressions: linear, exponential, potential, and logarithmic. The best $R^2$ value obtained was 0.62 for potential in Casanare. The values for the other departments were lower, which rendered the correlation invalid. Other regressions, carried out with average travel speed as the only independent variable, produced $R^2$ values that did not register above 0.32, which led us to discarded the theory that the productive condition of municipalities in their respective departments was related to the average travel speed to and from their villages and districts; the variable can be interpreted as the average condition of the roadway infrastructure (Tables 1 and 2).

**Table 1.** $R^2$ value when correlating MRW with the length of tertiary roadway network between villages and districts with their municipal capital.

| Department | Linear | Logarithmic | Exponential | Potential |
|---|---|---|---|---|
| Casanare | 0.12 | 0.44 | 0.16 | 0.62 |
| Cundinamarca | 0.003 | 0.007 | 0.003 | 0.016 |
| Meta | 0.007 | 0.023 | 0.003 | 0.04 |
| Santander | 0.37 | 0.31 | 0.04 | 0.24 |
| Quindío | 0.016 | 0.19 | 0.028 | 0.22 |

**Table 2.** $R^2$ value correlates MRW and average travel speed between villages and districts with their municipal capital.

| Department | Linear | Logarithmic | Exponential | Potential |
|---|---|---|---|---|
| Casanare | 0.01 | 0.05 | 0.03 | 0.002 |
| Cundinamarca | 0.004 | 0.0015 | 0.003 | 0.0006 |
| Meta | 0.029 | 0.011 | 0.02 | 0.019 |

Now, upon analyzing the previous correlations with R (Pearson correlation coefficient), the highest value obtained was 0.61 (including positive and negative), for the correlation between MRW and the length of the tertiary roadway networks between villages and districts with their municipal capital, for the Santander Department. However, the next highest value was 0.36 for the Casanare Department for the correlation of the same variables. The values are presented in Table 3. The validated coefficients sign is positive, for both R and $R^2$. The present analysis confirms the previous conclusions about correlations based on $R^2$ values.

**Table 3.** R value correlates MRW and length of tertiary roadway network and the average travel speed between villages and districts with their municipal capital.

| | Correlation between MRW and Length of Tertiary Roadway Network between Villages and Districts with the Their Municipal Capital | Correlation between MRW and Average Travel Speed between Villages and Districts with the Their Municipal Capital |
|---|---|---|
| Casanare | 0.36 | 0.23 |
| Cundinamarca | −0.06 | −0.07 |
| Meta | −0.09 | −0.17 |
| Santander | 0.61 | 0.21 |
| Quindío | 0.13 | 0.39 |

As such, an analysis was carried out for the five departments (adding the values of the municipalities of each department). In this case, changes in geographical context (changes from municipality to department) cause the dependent variable for indicating economic condition to not be MRW, but rather, Departmental Gross Domestic Product (GDP) (DANE, 2020) [28]. The independent variable was tertiary-level roadway network length between the municipalities and their villages and districts. As such, upon conducting linear, exponential, potential, and logarithmic regressions for the same dependent and independent variables, none of the resulting $R^2$ values approached a minimum of 0.7, ratifying that there is no valid relationship between the global departmental GDP, roadway length, and travel time in a roadway that connects its respective villages and districts (Table 4).

**Table 4.** $R^2$ value correlates GDP and tertiary roadway network length between villages and districts with its municipal capital.

|  | Linear | Logarithmic | Exponential | Potential |
|---|---|---|---|---|
| 5 Departments | 0.0005 | 0.0078 | 0.35 | 0.41 |

The second independent variable analyzed regarding departmental GDP was the average travel speed on routes between each municipality, their villages, and their districts. In this case, the hypothesis is the following: with greater travel speed, the municipality should have a greater development capacity (indicated by MRW) given that travel times are lower. Upon conducting correlations for linear, exponential potential, and logarithmic regressions, the greatest $R^2$ value obtained was 0.25 using linear regression (Table 5). These values led us to discard the correlation between the economic condition of each of the five analyzed departments and the average travel speed between several of its municipalities towards and from their respective villages and districts.

**Table 5.** $R^2$ value when correlating departmental GDP and average travel speed between villages and districts with their municipal capital.

|  | Linear | Logarithmic | Exponential | Potential |
|---|---|---|---|---|
| 5 Departments | 0.25 | 0.24 | 0.23 | 0.22 |

Until this moment, the variables of travel between each department's municipality and their connecting villages have not been indicators of the economic level within the region. Given that the relation of MRW did not result in viable travel between each municipality and its villages and districts, we proceeded to analyze the correlation of MRW by adding travel time and distance between each municipality and the capital city of each department. The correlation result for these five independent variables (average travel distance and speed between each municipality and the capital city of each department) resulted in a high-value adjusted $R^2$ for Casanare and Quindio (0.73 and 0.97, respectively), but with very low values for Cundinamarca, Meta, and Santander (0.25, 0.30, and 0.36, respectively). These last numbers do not grant statistical validity to the correlation. Additionally, the signs of coefficients are not equal in the five studied departments. Therefore, we establish that the economic productivity condition of a municipality (regarding the department it belongs to, and measured through MRW) is related neither to the length of the roadway network which interconnects its villages and districts, nor to the average travel time that takes place in these routes.

As such, the variable speed was eliminated, given that in the analysis above, it was observed that a multicollinearity condition occurred because of the resulting velocity of the direct relationship between travel time and distance. Despite this, the adjusted $R^2$ values were low: Casanare (0.32), Cundinamarca (0.17), Meta (0.10), Santander (0.37) y Quindío (0.16). Then, correlations were carried out for all the studied municipalities, without discriminating them by department. In total, these add up to 247 municipalities. The results also denied a statistical correlation. Using the total kilometer distance to villages and districts and the average travel speed in these trajectories, and MRW as a dependent variable, the adjusted $R^2$ was only 0.043. The result was similar when correlating MRW only with the kilometer distance, like when using the average speed (0.001). Given the above, this analysis led us to discard the correlation between the municipal networks connecting to villages and districts and the economic position of each municipality. In a complementary manner, the following regressions were carried out: linear, potential, exponential, and logarithmic; however, the $R^2$ values varied between 0.003 and 0.12 for the analysis using the length of the tertiary network (Table 6), and between 0.0038 and 0.0074 for the analysis that used average travel speed (Table 7). With the above, it can be ratified that the MRW of municipalities in Colombia does not correlate with the length of the roadway network

which connects to its villages and districts, nor with average travel speed on these routes or sections.

**Table 6.** $R^2$ value upon correlating MRW and the tertiary roadway length between villages and districts with their municipal capital (247 municipalities).

|  | **Linear** | **Logarithmic** | **Exponential** | **Potential** |
|---|---|---|---|---|
| 247 municipalities | 0.044 | 0.119 | 0.003 | 0.046 |

**Table 7.** $R^2$ value upon correlating MRW and average travel speed between villages and districts with their municipal capital (247 municipalities).

|  | **Linear** | **Logarithmic** | **Exponential** | **Potential** |
|---|---|---|---|---|
| 247 municipalities | 0.006 | 0.006 | 0.0074 | 0.0038 |

To continue the analysis of correlations, we proceeded to only work with travel conditions between municipalities and the capital of their respective department. Linear, exponential, potential, and logarithmic regressions were carried out between the MRW of each municipality and the average travel time to the capital city (recalling that this variable consists of times georeferenced by Google Maps on a workday and during peak hours). The $R^2$ for the department of Quindio varied between 0.40 and 0.97, for Casanare between 0.21 and 0.70, for Cundinamarca between 0.17 and 0.33, for Meta between 0.026 and 0.29, and for Santander between 0.0004 and 0.025 (Figure 1).

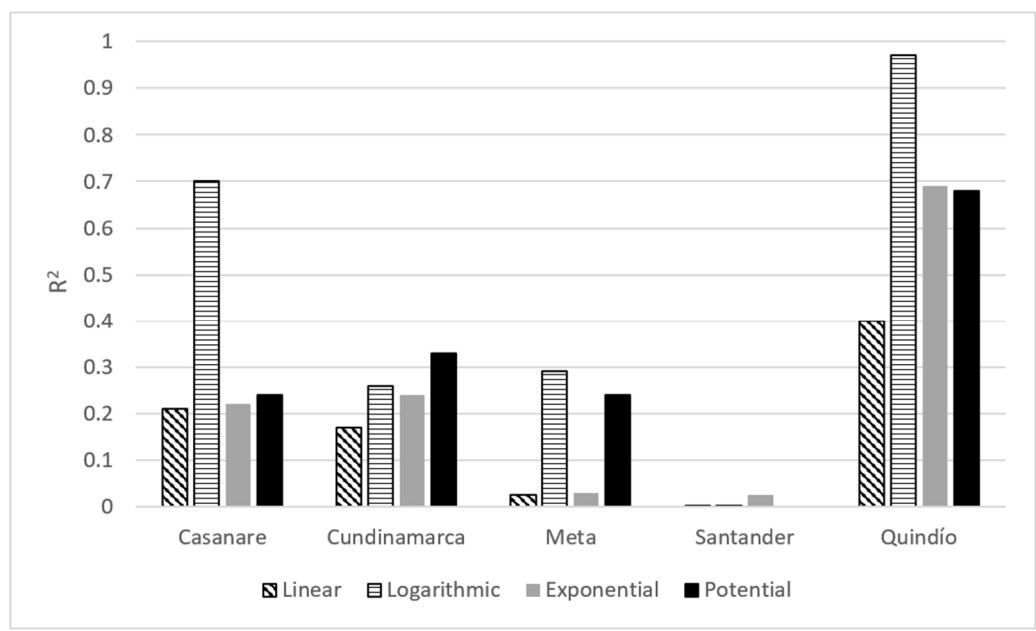

**Figure 1.** $R^2$ of regressions between MRW and average travel time between various municipalities and the capital city of their department.

Given that the average speed in tertiary roadways is usually very similar throughout the entire national territory, when changing the independent variable for travel distance to the capital, the $R^2$ values were similar to those analyzed with the average times: for Quindio between 0.39 and 0.95, for Casanare between 0.16 and 0.70, for Cundinamarca between 0.19 and 0.37, for Meta between 0.008 and 0.25, and for Santander between 0.0002 and 0.017. With these correlations, both with the time and distance between municipalities and the capital city of their department, the relationship with a better economic position of municipalities on a departmental level was discarded given that only two departments (of the five analyzed) presented values greater than or equal to 0.70 (Figure 1). Therefore, there

is no significant relationship between the MRW of municipalities and the condition of their respective roadway travel connecting to their capital city.

### 3.2. Analysis According to Departmental GDP and Roadway Conditions

The previous section reports the results of correlating the GDP of five departments (for which information was obtained in-field in regard to times, distances, and speeds) with the average conditions of kilometer distance and time between municipalities and their respective villages and districts. The results only validated a correlation for two departments. Because of this, we decided to correlate the roadway network variables for 27 of the 32 departments in Colombia, to establish whether the departmental GDP is correlated with the total kilometer distance of paved roadways (with the following conditions: Very Good, Good, Regular, Poor, or Very Poor) or non-paved roads. The information regarding GDP was obtained from DANE (2020) [28], and the information regarding kilometer distance was obtained from INVIAS (2020) [32].

Using departmental GDP 2020 as the dependent variable and departmental kilometer-distance paved roadway length with its five condition levels as an independent variable (from Very Good to Very Poor), and after conducting exponential, potential, linear, and logarithmic regressions, the $R^2$ values were between 0.005 and 0.06 for the condition Very Good, between 0.05 and 0.18 for Good, between 0.05 and 0.28 for Regular, between 0.05 and 0.22 for Poor, and finally, between 0.008 and 0.11 for Very Poor. Then, the correlation results were studied, but with the non-paved roadway network as an independent variable. Once again, the following regressions were carried out: exponential, potential, linear, and logarithmic. The $R^2$ values were between 0.005 and 0.009 for the condition Very Good, between 0.0005 and 0.015 for Good, between 0.00001 and 0.01 for Regular, between 0.00009 and 0.018 for Bad, and finally, between 0.00001 and 0.04 for Very Poor. In both cases—paved and non-paved roadways—the $R^2$ values did not reach close enough to a minimum acceptable value of 0.7 to validate a correlation. Figure 2 shows that the $R^2$ values for non-paved roadways are strongly inferior to those of paved roadways; in other words, the relationship is somewhat greater between the departmental economy and a paved roadway in comparison to a non-paved roadway. In summary, there is no significant relationship between the global GDP of each department and roadway network length, both for paved and non-paved roadways.

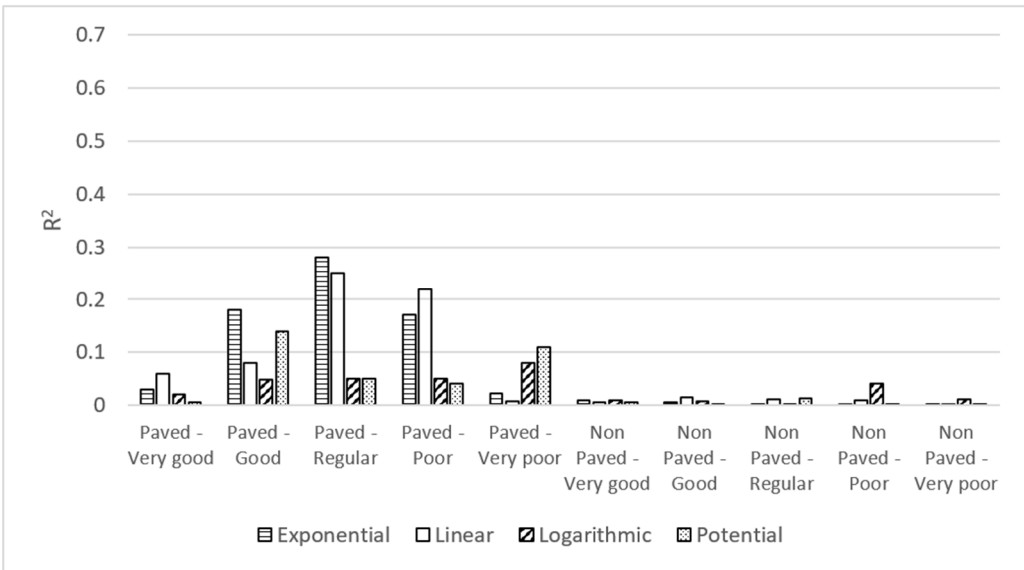

**Figure 2.** The $R^2$ value for regressions between departmental GDP and roadway network length according to condition.

Now, upon analyzing the previous correlations with R (Pearson correlation coefficient), the highest value obtained was 0.50 (including positive and negative) for the global GDP correlated with the extension of the paved–regular roads (Table 8). In summary, there is no significant relationship between the global GDP of each department and roadway network length, both for paved and non-paved roadways. The validated coefficients sign is positive, for both R and $R^2$.

**Table 8.** R value upon correlating global and sector GDP, and length of tertiary road network in different conditions.

|                        | Global GDP | Agrobusiness GDP | Mining GDP | Agrobusiness Plus Mining GDP |
|------------------------|------------|------------------|------------|------------------------------|
| Paved—Very good        | 0.26       | 0.22             | −0.10      | 0.07                         |
| Paved—Good             | 0.29       | 0.45             | 0.25       | 0.44                         |
| Paved—Regular          | 0.50       | 0.55             | 0.26       | 0.50                         |
| Paved—Poor             | 0.47       | 0.46             | 0.21       | 0.42                         |
| Paved—Very poor        | 0.09       | 0.13             | 0.04       | 0.10                         |
| Non-paved—Very good    | −0.07      | 0.05             | 0.55       | 0.38                         |
| Non-paved—Good         | −0.12      | −0.03            | 0.26       | 0.15                         |
| Non-paved—Regular      | −0.10      | 0.03             | 0.16       | 0.12                         |
| Non-paved—Poor         | −0.10      | 0.02             | 0.09       | 0.07                         |
| Non-paved—Very poor    | −0.05      | 0.05             | −0.13      | −0.05                        |

Regarding the coefficient signs in the regressions for multiple variables, as for the paved roadway network, these were positive. As for the non-paved roadway network, these were negative. The above also led us to discard appropriate statistical behavior, given the length in poor conditions (Poor and Very Poor) (negative sign). However, all the signs for the paved roadway network were positive. As for the non-paved roadway network, the effect was similar. In other words, with greater length of the network in good or regular condition, we would expect to find a greater GDP, and the opposite would be expected for roadways in poor condition; however, the sign in all the coefficients were negative. With the above, it is possible to conclude that there is a greater relationship between GDP and a greater length of paved roadway than that occurring between GDP and non-paved roadway length.

### 3.3. Analysis According to Rural Sector GDP and Roadway Conditions

Given that there was no correlation between the roadway network variables and departmental GDP, correlations were proposed, but only with the components that have a greater impact upon the rural sector: GDP 2020 for the agriculture, livestock, hunting, fishing, silviculture (agribusiness), and mining sectors. By adding these records as a dependent variable, we proceeded to carry out correlations with the kilometer distance of the paved roadway network in its different conditions (Very Good, Good, Regular, Poor, and Very Poor), and likewise with the non-paved roadway network. Starting the analysis with the paved roadway network and the sum of agribusiness and mining GDP, regressions were carried out for the following models: exponential, potential, linear, and logarithmic. The $R^2$ values were between 0.0003 and 0.025 for the condition Very Good, between 0.19 and 0.5 for Good, between 0.25 and 0.38 for Regular, between 0.15 and 0.20 for Poor, and finally, between 0.01 and 0.11 for Very poor. The analysis of the non-paved roadway network presents a greater error than the one for the paved roadway network. The greatest $R^2$ value is 0.14 in the condition Very Good, and there is a linear trend (Figure 3). In summary, the length of roadway network in departments, both for paved and non-paved roadway networks, independent of its condition or state, does not correlate with

economic productivity in the agricultural or mining sectors within the territory, represented in sector GDP.

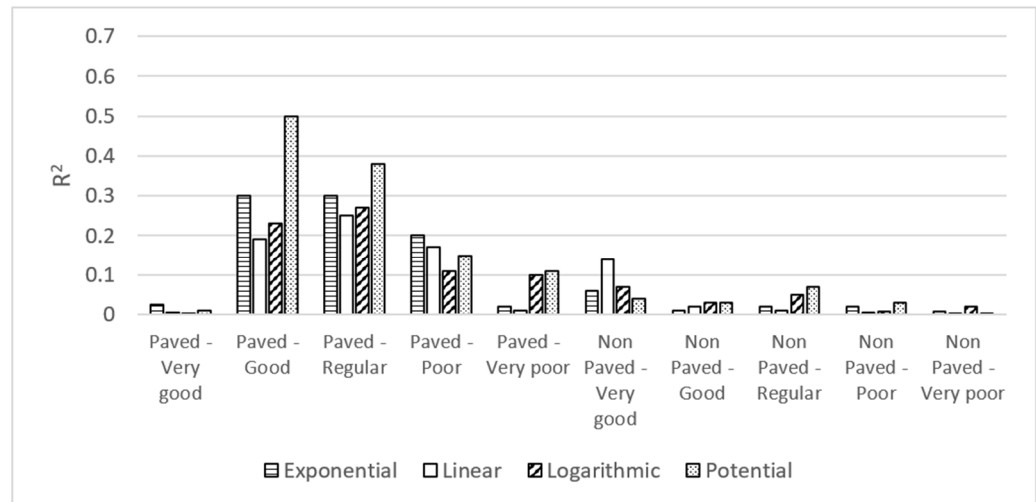

**Figure 3.** The $R^2$ value for regressions between departmental GDP for the agriculture, livestock, silviculture, hunting, fishing, and mining sectors, and roadway network length according to condition.

Given that the correlation between the agribusiness and mining GDP was not statistically valid, we decided to analyze only the agribusiness GDP, given that these are the activities with the greatest participation in the Colombian rural economy. Upon conducting these correlations, the greatest $R^2$ value for the linear, exponential, potential, and logarithmic regressions was 0.44 for the condition Good, and potential regression (Figure 4). For the correlation between agribusiness GDP and the non-paved roadway network, the resulting $R^2$ values for the linear, exponential, potential, and logarithmic regressions do not exceed the value of 0.03 (Figure 4).

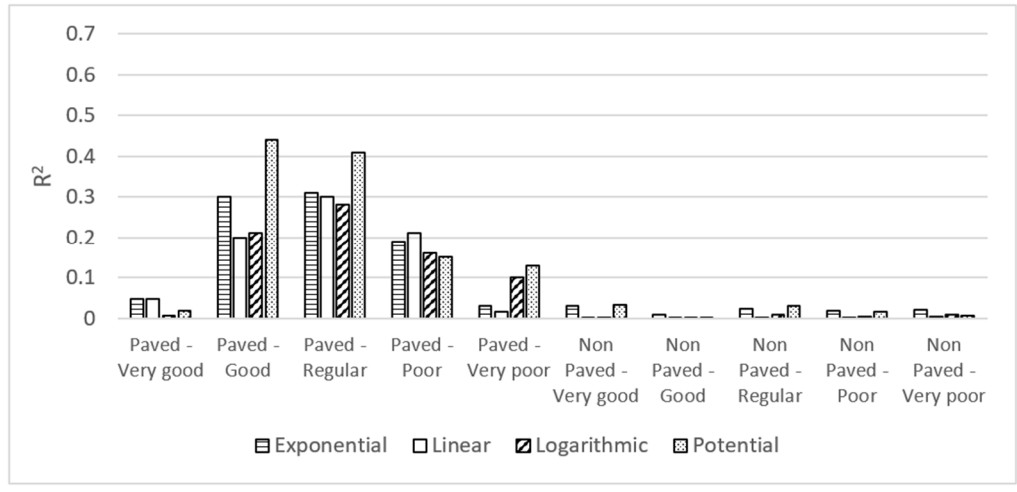

**Figure 4.** The $R^2$ value for regressions between departmental GDP and the agriculture, livestock, silviculture, hunting, and fishing sectors, and roadway network length according to condition.

As such, we decided to apply the correlation, but now with the average annual agribusiness GDP growth (agriculture, livestock, silviculture, hunting, and fishing) between the years 2005 and 2020. Likewise, there was no statistical validity obtained, given that the $R^2$ values were between 0.0017 and 0.042 for the condition Very Good, between 0.21 and 0.37 for Good, between 0.25 and 0.40 for Regular, less than 0.056 for Poor, and between 0.083 and 0.13 for Very Poor. For the total paved roadway network as an independent variable, potential regression produced an $R^2$ value of 0.37. This means that once again,

there is no significant statistical correlation between the tertiary roadway network length of a department and the increase or decrease in agribusiness GDP within the same territory.

Regarding the non-paved roadway, $R^2$ values with the greatest resulting value were for the conditions Very Good (0.17), Good (0.12), Regular (0.12), Poor (0.10), and Very Poor (0.033), and for the total non-paved-roadway network (0.12). Therefore, neither agribusiness, the agribusiness and mining GDP value, nor the annual rates for both variables display a trend that statistically validly correlates with departmental roadway network length, according to its state or condition.

Continuing the analysis, the highest R value obtained (including positive and negative) was 0.55 for the agribusiness GDP correlated with the extension of the paved–regular roads (Table 8). In summary, reviewing the R and $R^2$ values, there is no significant relationship between the economic rural sectors' GDP for each department and roadway network length, both for paved and non-paved roadways. The validated coefficients sign is positive, for both R and $R^2$.

*3.4. Analysis According to Departmental GDP and Roadway Network Classification*

The previous sections analyzed roadway condition (paved or non-paved with its 5 levels: Very Good, Good, Regular, Poor, or Very Poor). The study continues with the classification of the roadway network (primary, secondary, and tertiary). Among the first inquiries, we wanted to know if the roadway network length of the tertiary roadway network in each of the departments presents any correlation with the territory's territorial extension. Upon carrying out the correlation, with the information provided by the Ministry of Transport (2019), the $R^2$ value was 0.032, which led us to discard the hypothesis that correlates trend of the tertiary roadway network density occupying and the extension of the territory. In other words, with greater territorial extension, there should be greater roadway network length.

As such, we proceeded to correlate variables under the premise that with greater length of the tertiary roadway network of a department, the department will experience better economic performance (analyzing GDP). After conducting linear, exponential, potential, and logarithmic regressions between departmental GDP and tertiary roadway network length in each department, the obtained $R^2$ value was 0.28 for linear, 0.09 for exponential, 0.38 for logarithmic, and 0.40 for potential (Figure 5). A similar analysis was carried out, but using primary roadway network length, obtaining an $R^2$ value of 0.38 for linear, 0.50 for exponential, 0.24 for logarithmic, and 0.53 for potential. The greatest value (0.53) resulted in being less favorable, but was close to 0.56, resulting from the analysis between Latin American countries (Urazan et al. 2017) (Figure 5). Given the above, it is deduced that the roadway network length in departments does not adequately correlate with the general GDP in each territory.

The GDP of the departmental agribusiness sector was correlated with the tertiary network, with the following results for the $R^2$ values: 0.47 for exponential regression, 0.55 for linear, 0.18 for logarithmic, and 0.69 for potential (Figure 5). Upon adding Adding the mining GDP $R^2$ values to those of agrobusiness agribusiness GDP, the resulting regressions were as follows: 0.34 for exponential regression, 0.28 for linear, 0.12 for logarithmic, and 0.67 for potential (Figure 5). The $R^2$ values regarding the agribusiness sector (0.69) resulted in a greater value than when including the mining sector (0.67), which was very close to 0.7 and the reason for which the resulting regression was analyzed (Figure 6) Equation (1). Unlike what was concluded in the previous paragraph, the GDP of the agribusiness sector does present an acceptable correlation ($R^2$ value close to 0.7) with the tertiary roadway network length in each department. Additionally, the standard error results in a value of 0.09 and a *p*-value of $1.41 \times 10^{-5}$, confirming that there is a statistically significant correlation.

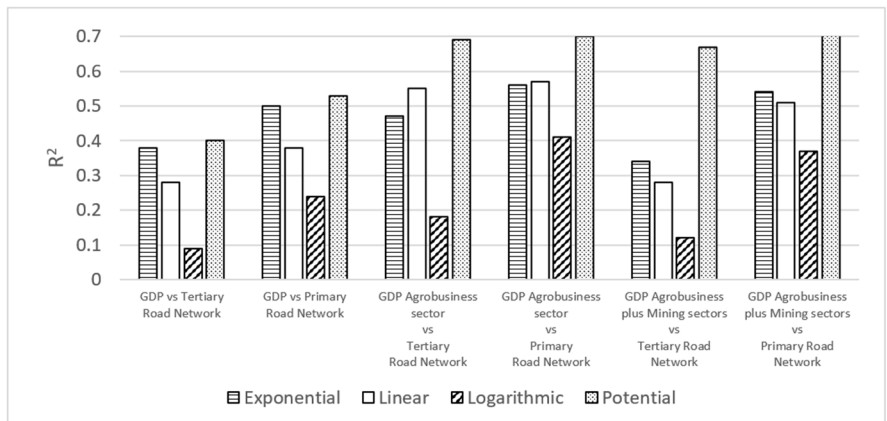

**Figure 5.** The $R^2$ value for regressions between departmental GDP, and departmental GDP for the agriculture, livestock, silviculture, hunting and fishing sectors, and for agriculture, livestock, silviculture, hunting, and fishing plus mining (dependent variables), with primary and tertiary roadway network length (independent variables).

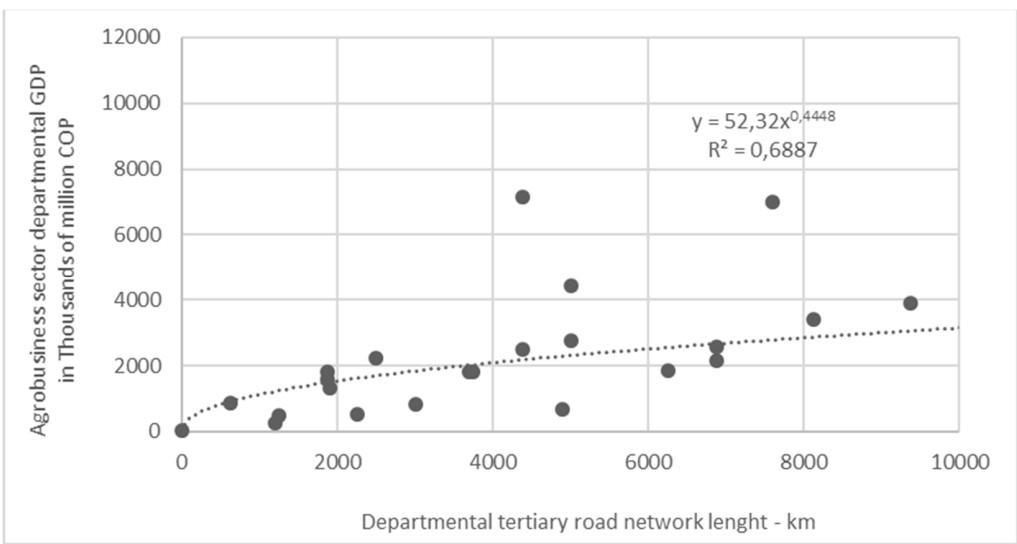

**Figure 6.** Potential regression between agribusiness GDP and tertiary roadway network length, according to the department.

The resulting Equation (1) reiterates the exponent's positive sign, given that with a greater tertiary roadway network length, there is a trend of greater GDP in the agribusiness sector. Only 2 departments of the 27 analyzed (7.5%) present a graphical position that can be considered as relatively distant from the trend line: Santander (7600, 6985) and Valle (4375, 7142); these are the two departments with the greatest GDP in the agribusiness sector.

$$y = 52.32 \times x^{0.4448} \tag{1}$$

where $y$ = departmental GDP in the agribusiness sector (thousands of millions COP) and $x$ = the departmental tertiary roadway network (km).

Similarly, the primary roadway network length was correlated with agribusiness sector GDP, obtaining $R^2$ values of 0.56 for exponential, 0.57 for linear, 0.41 for logarithmic, and 0.70 for potential (Figure 5). Upon adding mining GDP to agribusiness GDP, the $R^2$ values were 0.54 for exponential regression, 0.51 for linear, 0.37 for logarithmic, and 0.73 for potential (Figure 5). The values of the two potential regressions were greater than or equal to 0.7, providing statistical validity to the correlations. Given the above, we proceeded to analyze the results of the regressions (Figures 7 and 8; Ecs. 2 and 3),

obtaining a good $R^2$ value between the primary roadway network of each department and the agribusiness sector GDP in that same territory. Additionally, the standard error was 0.94 (higher than with a tertiary road network) and the *p*-value was $6.5 \times 10^{-6}$, which confirms a valid statistical correlation.

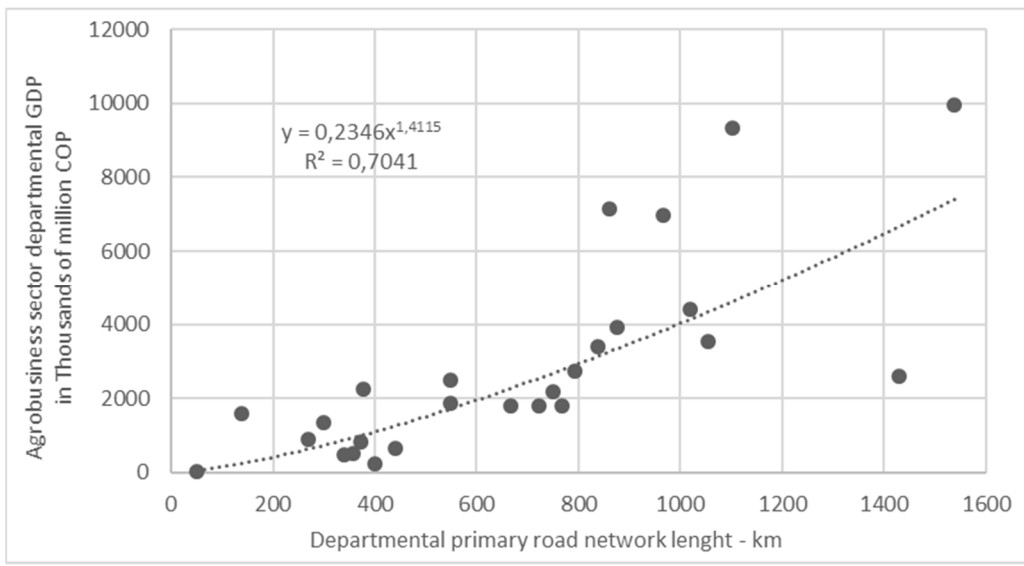

**Figure 7.** Potential regression between agribusiness sector GDP and primary roadway network length according to the department.

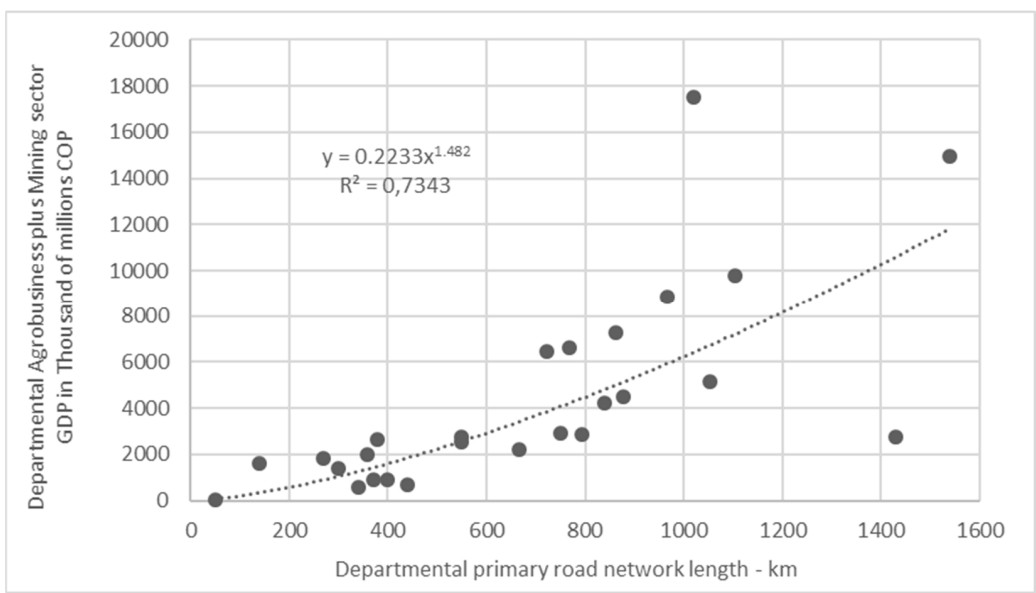

**Figure 8.** Potential regression between agribusiness plus mining sector GDP and primary roadway network, according to the department.

The equation resulting from Figure 7, Equation (2), reiterates the exponent's positive sign, given that with a greater primary roadway network length, there is a trend of obtaining greater GDP in the agribusiness sector. In this case, it is possible to consider that 4 departments (the greatest agribusiness sector GDP) out of the 27 analyzed (15%) are relatively distant from the trend line: Antioquia (1539, 9948), Cundinamarca (1103, 9350), Santander (966, 6985), and Valle (862, 7142).

$$y = 0.2346 \times x^{1.4115} \tag{2}$$

where $y$ = departmental agribusiness sector GDP (thousands of millions COP) and $x$ = the departmental primary roadway network (km).

From the regression that includes mining sector GDP with the primary roadway network, Figure 8 and Equation (3) are obtained. In this case, Equation (3) also reiterates the exponent's positive sign, given that with greater primary roadway network length, there is a trend of obtaining greater GDP in the agribusiness + mining sector. In this case, it is possible to consider that 3 departments out of the 27 analyzed (11%) are relatively distant from the trend line: Antioquia (1539, 14380), Meta (1019, 17502), and Cauca (1429, 2273). The greatest $R^2$ value (0.73) results from correlating departmental agribusiness sector + mining sector GDP with the primary roadway network, which means that primary roadway network length contributes more to the mining, agribusiness, livestock, hunting, silviculture, fishing, and mining sectors in different territories or departments within the country in comparison to tertiary roadway network length, despite the latter allowing access to rural zones. In the case of the primary roadway network, the analysis results in a standard error of 1.64 (higher than with only agribusiness GDP) and a *p*-value of $4.08 \times 10^{-5}$.

$$y = 0.2233 \times x^{1.482} \tag{3}$$

where $y$ = PIB departmental agribusiness sector GDP + mining sector GDP (thousands of millions COP) and $x$ = the departmental primary roadway network (km).

Expanding the analysis, the statistically significant R values (including positive and negative) are 0.76 for the primary road network correlated with agribusiness sector GDP, 0.74 for the tertiary road network correlated with agribusiness sector GDP, and 0.71 for the primary road network correlated with agribusiness plus mining sector GDP (Table 9). These three cases coincide with those that were validated using the $R^2$ values and maintain the correlations with a positive sign.

**Table 9.** R value upon correlating global and sector GDP with length of the primary and tertiary road network by department.

|                      | Global GDP | Agrobusiness GDP | Mining GDP | Agrobusiness Plus Mining GDP |
| -------------------- | ---------- | ---------------- | ---------- | ---------------------------- |
| Primary Road Network | 0.62       | 0.76             | 0.38       | 0.71                         |
| Tertiary Road Network | 0.53      | 0.74             | 0.12       | 0.53                         |

At the end of the analysis, the extension of the tertiary road network was correlated with the GDP of the two economic sectors that have the greatest participation: commerce (17%) and the manufacturing industry (13%). The agricultural sector added to the mining sector contributed 11% (annual average from 2005 to 2020) [33].

In a similar manner to the previous analysis, tertiary roadway network length was correlated with the commerce sector GDP, obtaining $R^2$ values of 0.18 for exponential regression, 0.23 for linear, 0.34 for logarithmic, and 0.22 for potential (Figure 9). The best result (the linear case) corresponds with Equation (4). Upon adding manufacturing industrial GDP to commerce GDP, the $R^2$ values were 0.20 for exponential regression, 0.26 for linear, 0.38 for logarithmic, and 0.27 for potential (Figure 10). In this case, the best result (the logarithmic case) corresponds to Equation (5). The values were lower than 0.7, which led us to discard the trend with correlations. Therefore, it is concluded that the development of the tertiary road network in Colombia is related to rural activities and not to the overall regional departmental economy. However, these two equations are not relevant because $R^2$ is statistically inappropriate, value less than at much lower than 0.7.

$$y = 2315.2 \ln x - 13269 \tag{4}$$

where $y$ = PIB departmental commerce sector GDP (thousands of millions COP) and $x$ = the departmental tertiary roadway network (km).

$$y = 1997.9 \ln x - 11511 \tag{5}$$

where $y$ = PIB departmental commerce plus manufacturing sector GDP (thousands of millions COP) and $x$ = the departmental tertiary roadway network (km).

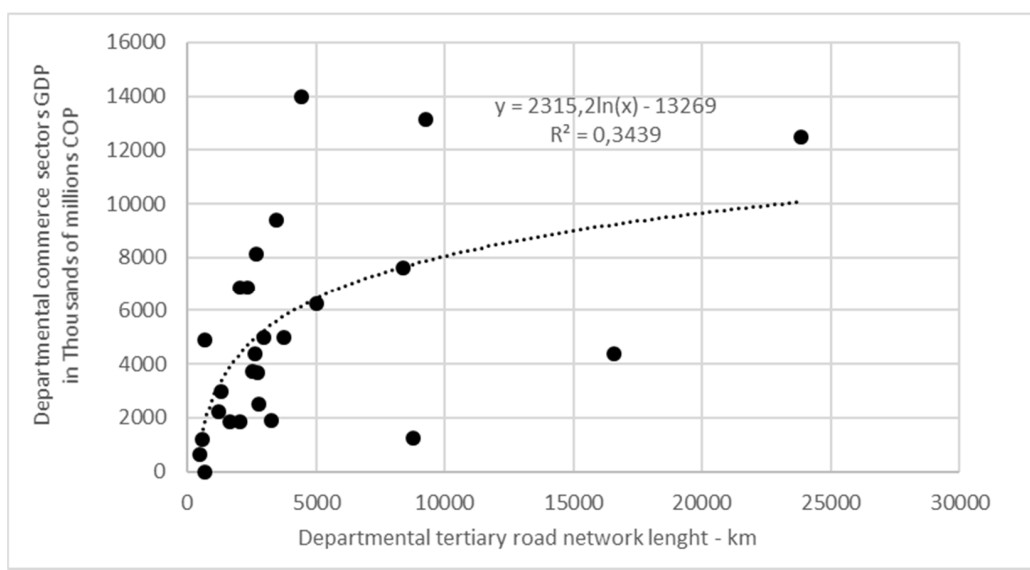

**Figure 9.** Logarithmic regression between commerce sector GDP and tertiary roadway network, according to the department.

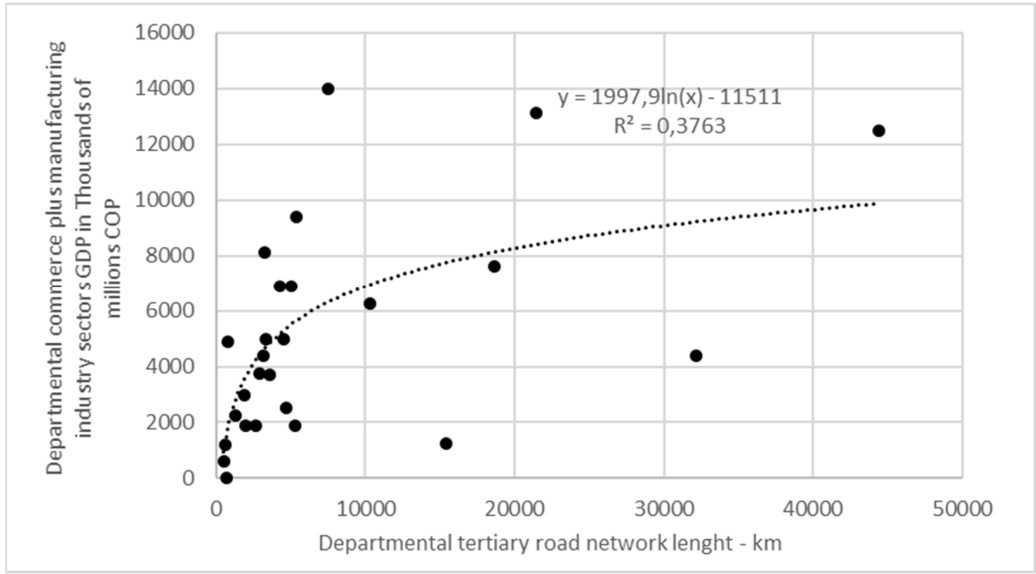

**Figure 10.** Logarithmic regression between commerce plus manufacturing industry sector GDP and tertiary roadway network, according to the department.

## 4. Discussion

There are not many studies that analyze the correlation between the quantity and condition of the roadway networks of a municipality or department with variables that measure the economic level within a territory. Aiming at this point, this paper contributes to the state of the art, by attempting to correlate travel conditions (time, average speed)

and roadway length with economic development indicators such as departmental Gross Domestic Product and Municipal Relative Weight for municipalities.

In the case of Colombia, and probably in other developing countries, the tertiary road network is the largest in the entire national territory. Between 2005 and 2020, it has grown from 62% to 69% of the total road network and its length has gone from 75,000 to 142,000 km. Additionally, it must be highlighted that 71% of the tertiary road network is administrated with municipal resources [34].

The growing proportion of the tertiary road network is related to a greater contribution to the economy of rural areas because they are the regions to which it provides accessibility. By their very definition, rural areas are tied to agricultural, livestock, and mining economic activities (mainly) and need a connection to roads for their development. The participation of agriculture and mining in the national annual GDP has averaged 11% since 2005 [33], but in rural areas, they are the most important economic sectors. The commerce, education, health, and services sectors (with greater participation in the national GDP) serve the majority of the population in the main urban areas (77% in the Colombian case) [35]. It must even be said that the manufacturing industry and commerce sectors (13% and 17% of Colombian GDP) do not represent a valid correlation with the tertiary road network. The highest $R^2$ value is 0.38.

The results obtained in this paper were developed from Pearson and Spearman correlation analyses ($R^2$ and *p*-value). These methodologies are commonly used to validate, or not, the hypothetically proposed correlations. The data show that the management of the tertiary road network depends mainly on rural economic activities and not global economic performance. This conclusion allows future research to focus on the improvements that must be made so that the economic sectors in the country help improve the accessibility and quality of life of the inhabitants of the most distant regions through the tertiary road network.

It is important to mention that if there is a correlation between two variables, it does not imply that the independent variable explains the behavior of the dependent variables; that is, there is not necessarily a causal relationship between the two variables. Therefore, the explanations of the relationships between variables, exposed in this discussion, stem from social and economic principles and the state of the art, and do not pretend to be supported only by validated statistical correlations.

The study intends that the relationships be thoroughly statistically validated to identify the indicators that require more detailed subsequent studies to establish investment policies in highways that have an economic impact on rural areas.

As previously mentioned, this paper aims to contribute significantly to the state of the art, especially to the infrastructure scenario in Latin America; it ratifies that the development of the rural economy requires adequate extension of the tertiary road network in regions, but not so much for global economies, because the highest population density is concentrated in urban areas.

Another aspect to highlight is that the spatial density of the road network is not similar in all regions; therefore, territorial extension does not correlate with rural economic development as the extension of the tertiary road network does.

Linked to the above, there was no statistical accuracy in the values of $R^2$ resulting from the correlation between the travel condition (average speed and distance) from each municipality to the department capital city, with the economic level (GDP) of those same territorial zones.

Finally, the present study clarifies several beliefs or hypotheses frequently used regarding the economic and development variables that have a high correlation with the condition of the network of the tertiary or rural roads in a territory, at least in the Colombia case, and probably in other Latin American countries.

## 5. Conclusions

This research contributes to the state of knowledge about the relationship between investment in the road network and economic growth in Latin American countries, specifically in the tertiary and rural road networks; this is a subject of great interest again, because agricultural development has become essential for the region's economy.

Based on the results obtained in this study, the conclusions are as follows:

(1) None of the following premises had statistical validity upon being applied to Colombian municipalities: (i) having a greater roadway network length in rural areas or better travel conditions, measured by average speed, correlates with a better conditions in economic sectors; (ii) fewer distances and greater average travel speeds between a department's municipalities and its capital city are correlated with better economic conditions for the municipalities within that department; and (iii) departments with greater roadway network length (paved or non-paved), and especially those in good condition, are correlated with better economic conditions.

(2) Regressions between departmental GDP (dependent variable) and roadway network length, both primary and tertiary (independent variable), do not comply with a statistically valid $R^2$. However, regressions that used agribusiness sector GDP (agriculture, livestock, hunting, fishing, and silviculture) with tertiary roadway network length as an independent variable presented an $R^2$ of 0.69, a number which validates the relationship between the variables. A similar situation was recorded with the primary roadway network in each department as the independent variable ($R^2 = 0.70$). This is interpreted as an important relationship between a department's economic development (global GDP) and production in the agribusiness sector.

(3) Mining sector GDP, which makes up part of the development in rural zones, presented a better correlation with the primary roadway network ($R^2 = 0.73$). This is the greatest correlation in the entire study. This allows us to infer that at a departmental level, there is a relationship between the length of a primary roadway network and mining sector production in the respective region. This relationship is more statistically important than the relationship in the agribusiness sector.

(4) The level of development in a municipality, compared to other municipalities in its department, does not correlate with the distance and time from its villages and districts on the roadway.

(5) If the relationship between the tertiary rural network and economic development in the country's departments (measuring the latter in general GDP) is the same, there is no valid correlation.

(6) However, if the analysis is carried out concerning the GDP of the agribusiness and mining sectors, a statistically valid correlation is obtained ($R^2$ equal to or greater than 0.7). Additionally, upon correlating sector GDP with the primary roadway network, the resulting $R^2$ value is 0.7 for the primary network and drops to 0.67 for the tertiary roadway network.

(7) Despite what was expressed in the paragraph above, if the mining sector is eliminated from the analysis and only the agribusiness component is left, the resulting $R^2$ value is 0.7 both for primary and tertiary roadway networks; thus, we conclude that the activities in rural areas are those which most impact investment in the roadway network for a region, in the case of Colombian departments. This explains that the greatest economic productivity of departments is in their respective capital cities, and the need to extend the roadway network (especially tertiary or rural) is to connect to other municipalities whose productive trend is of the agribusiness/mining type. In other words, if there were no municipalities with agribusiness productivity in their departments, it would not be necessary to extend their roadway network; moreover, the primary network would predominate among department capitals, and it would not be important for tertiary roadway network to be extended.

(8) Additionally, the obtained results establish, for the case of Colombia, that there is a valid correlation between the length of a primary and tertiary roadway network

and the GDP in the country's departments; this ratifies the published study's main conclusion, according to which the GDPs of the main countries in Latin America and the Caribbean are correlated with the length of their corresponding primary roadway network.

**Author Contributions:** Conceptualization, C.F.U.-B. and M.A.C.-L.; methodology, C.F.U.-B.; software, H.A.R.-Q.; validation, C.F.U.-B., M.A.C.-L. and H.A.R.-Q.; formal analysis, H.A.R.-Q.; investigation, C.F.U.-B. and M.A.C.-L.; resources, C.F.U.-B., M.A.C.-L. and H.A.R.-Q.; data curation, H.A.R.-Q.; writing—original draft preparation, C.F.U.-B.; writing—review and editing, M.A.C.-L.; visualization, C.F.U.-B.; supervision, H.A.R.-Q.; project administration, C.F.U.-B., M.A.C.-L. and H.A.R.-Q.; funding acquisition, C.F.U.-B., M.A.C.-L. and H.A.R.-Q. All authors have read and agreed to the published version of the manuscript.

**Funding:** This research received no external funding.

**Institutional Review Board Statement:** Not applicable.

**Informed Consent Statement:** Not applicable.

**Data Availability Statement:** Not applicable.

**Conflicts of Interest:** The authors declare no conflict of interest.

## Appendix A

**Table A1.** Casanare department municipalities studied information: MRW, tertiary road network length, and average travel speed.

| Casanare Department Municipalities Studied | Municipal Relative Weight (MRW) (%) 2019 | Tertiary Road Network Analyzed (km) | Tertiary Road Network Average Travel Speed (km/h) | Average Travel Speed (km/h) to Department Capital City |
|---|---|---|---|---|
| Aguazul | 11.7 | 214.7 | 22 | 41.54 |
| Chámeza | 0.3 | 48.6 | 20 | 38.40 |
| Hato Corozal | 1.3 | 411.5 | 21 | 61.86 |
| La Salina | 0.1 | 34.1 | 22 | 45.41 |
| Maní | 3.8 | 409.8 | 15 | 52.26 |
| Monterrey | 1.5 | 165.7 | 22 | 44.64 |
| Nunchía | 1.5 | 183.8 | 23 | 46.96 |
| Oracle | 5.5 | 454.7 | 15 | 46.61 |
| Paz de Ariporo | 5.3 | 566.9 | 24 | 56.91 |
| Póre | 1.4 | 161.3 | 17 | 54.68 |
| Receptor | 0.1 | 67.6 | 16 | 39.79 |
| Sabana Larga | 0.4 | 90.5 | 18 | 46.74 |
| San Luis de Palenque | 2.9 | 405.3 | 19 | 60.00 |
| Támara | 0.6 | 120.1 | 11 | 51.11 |
| Tauramena | 16.2 | 265.7 | 20 | 43.33 |
| Trinidad | 2.6 | 315.6 | 16 | 58.43 |
| Villanueva | 11.7 | 190.3 | 25 | 47.79 |
| Yopal | 33 | 412.9 | 20 | 0.00 |

**Table A2.** Cundinamarca department municipalities studied information: MRW, tertiary road network length, and average travel speed.

| Cundinamarca Department Municipalities Studied | Municipal Relative Weight (MRW) (%) 2019 | Tertiary Road Network Analyzed (km) | Tertiary Road Network Average Travel Speed (km/h) | Average Travel Speed (km/h) to Department Capital City |
|---|---|---|---|---|
| Agua de Dios | 0.2 | 127 | 16 | 41.33 |
| Alban | 0.2 | 73.4 | 17 | 36.00 |
| Anapoima | 0.9 | 311.2 | 17 | 39.71 |
| Anolaima | 0.3 | 110.5 | 17 | 61.64 |
| Apulo | 0.2 | 92.6 | 18 | 41.22 |
| Arbeláez | 0.4 | 144.5 | 16 | 35.29 |
| Beltrán | 0.1 | 25.1 | 21 | 40.09 |
| Bituimá | 0.1 | 41.1 | 20 | 37.12 |
| Bojacá | 0.3 | 67.1 | 18 | 36.99 |
| Cabrera | 0.3 | 287.5 | 19 | 37.50 |
| Cachipay | 0.4 | 44.4 | 14 | 33.50 |
| Cajicá | 3 | 23.8 | 18 | 41.86 |
| Caparrapi | 0.3 | 373.1 | 22 | 42.00 |
| Cáqueza | 0.7 | 160.7 | 18 | 31.02 |
| Carmen de Carupá | 0.2 | 171 | 19 | 39.45 |
| Chaguaní | 0.1 | 87 | 14 | 34.95 |
| Chía | 5.6 | 38.3 | 18 | 32.09 |
| Chipaque | 0.1 | 171.7 | 20 | 30.33 |
| Choachí | 0.3 | 231.4 | 16 | 36.82 |
| Chocontá | 0.7 | 104.1 | 21 | 59.20 |
| Cogua | 1.1 | 141.6 | 17 | 44.79 |
| Cota | 5.3 | 21.7 | 15 | 25.45 |
| Cucunuba | 0.4 | 105.3 | 19 | 40.91 |
| El Colegio | 0.7 | 87.4 | 15 | 33.00 |
| El Peñón | 0.1 | 18.8 | 17 | 37.37 |
| El Rosal | 0.5 | 53.9 | 21 | 34.91 |
| Facatativá | 5.5 | 132.3 | 16 | 33.85 |
| Fomeque | 0.6 | 428.7 | 15 | 27.19 |
| Fosca | 0.2 | 150.6 | 17 | 30.39 |
| Funza | 5.2 | 20 | 20 | 26.18 |
| Fúquene | 0.2 | 58.1 | 21 | 43.48 |
| Fusagasugá | 3.4 | 334.7 | 16 | 36.48 |
| Gachalá | 0.1 | 75 | 25 | 36.61 |
| Gachancipá | 0.4 | 39.9 | 19 | 54.89 |
| Gacheta | 0.4 | 35.9 | 17 | 43.97 |
| Girardot | 2.9 | 26 | 19 | 43.52 |
| Granada | 0.2 | 20.4 | 16 | 33.49 |
| Guachetá | 0.2 | 198.3 | 17 | 38.85 |
| Guaduas | 0.7 | 91.2 | 20 | 40.22 |
| Guasca | 0.3 | 421.6 | 18 | 48.49 |
| Guataquí | 0.1 | 11.1 | 22 | 49.04 |
| Guatavita | 0.1 | 140.1 | 16 | 55.38 |
| Guayabal de Síquima | 0.2 | 32 | 19 | 34.29 |

**Table A2.** *Cont.*

| Cundinamarca Department Municipalities Studied | Municipal Relative Weight (MRW) (%) 2019 | Tertiary Road Network Analyzed (km) | Tertiary Road Network Average Travel Speed (km/h) | Average Travel Speed (km/h) to Department Capital City |
|---|---|---|---|---|
| Guayabetal | 0.1 | 79 | 16 | 21.73 |
| Gutiérrez | 0.1 | 138.7 | 17 | 26.41 |
| Jerusalén | 0.1 | 39.8 | 20 | 40.94 |
| Junín | 0.2 | 303.9 | 16 | 43.08 |
| La Calera | 0.9 | 171.6 | 16 | 28.80 |
| La Mesa | 2.1 | 170.4 | 15 | 37.76 |
| La Palma | 0.2 | 66.5 | 22 | 35.32 |
| La Pena | 0.1 | 43.7 | 17 | 36.45 |
| La Vega | 0.5 | 127.2 | 19 | 43.33 |
| Lenguazaque | 0.5 | 127.7 | 20 | 49.46 |
| Machetá | 0.1 | 31.7 | 19 | 55.33 |
| Madrid | 3 | 109.1 | 16 | 27.27 |
| Manta | 0.1 | 18.9 | 17 | 48.75 |
| Medina | 0.2 | 130.4 | 16 | 37.05 |
| Mosquera | 4.5 | 61.7 | 14 | 27.27 |
| Nariño | 0.1 | 18.5 | 13 | 47.18 |
| Nemocón | 0.2 | 78 | 20 | 44.44 |
| Nilo | 0.3 | 129.2 | 15 | 43.12 |
| Nocaima | 0.2 | 14.7 | 14 | 43.02 |
| Pacho | 0.6 | 156.6 | 19 | 39.34 |
| Paime | 0.1 | 113.8 | 18 | 33.19 |
| Pandi | 0.2 | 115.5 | 18 | 39.88 |
| Paratebueno | 0.5 | 187.3 | 20 | 36.50 |
| Pasca | 0.3 | 100.1 | 18 | 34.86 |
| Puerto Salgar | 0.5 | 118.1 | 24 | 47.34 |
| Pulí | 0.1 | 90.8 | 20 | 35.15 |
| Quebradanegra | 0.1 | 48.8 | 18 | 41.48 |
| Quetame | 0.2 | 80.6 | 18 | 35.73 |
| Quipile | 0.3 | 37.8 | 25 | 32.83 |
| Ricaurte | 0.5 | 44.5 | 19 | 43.40 |
| San Antonio del Tequendama | 0.3 | 38.9 | 17 | 33.64 |
| San Bernardo | 0.6 | 103.8 | 21 | 32.86 |
| San Cayetano | 0.2 | 60 | 24 | 29.86 |
| San Francisco de | 0.3 | 28.2 | 16 | 38.71 |
| San Juan de Rioseco | 0.1 | 44.7 | 19 | 39.04 |
| Sasaima | 0.4 | 81.8 | 14 | 38.10 |
| Sesquilé | 0.6 | 60.4 | 21 | 57.00 |
| Sibaté | 1.6 | 62.1 | 19 | 29.21 |
| Silvania | 0.6 | 33.8 | 15 | 57.43 |
| Simijaca | 0.3 | 37.7 | 21 | 46.54 |
| Soacha | 10.3 | 52 | 18 | 25.00 |
| Sopó | 1.6 | 52.6 | 20 | 49.33 |
| Subachoque | 0.3 | 77.2 | 20 | 38.10 |
| Suesca | 0.5 | 91 | 17 | 52.39 |

**Table A2.** *Cont.*

| Cundinamarca Department Municipalities Studied | Municipal Relative Weight (MRW) (%) 2019 | Tertiary Road Network Analyzed (km) | Tertiary Road Network Average Travel Speed (km/h) | Average Travel Speed (km/h) to Department Capital City |
|---|---|---|---|---|
| Susa | 0.1 | 43.2 | 18 | 46.85 |
| Sutatausa | 0.3 | 25.2 | 18 | 45.31 |
| Tabio | 0.4 | 76.1 | 17 | 36.00 |
| Tausa | 1 | 72.6 | 21 | 46.21 |
| Tena | 0.3 | 40 | 16 | 33.82 |
| Tenjo | 2.2 | 69 | 17 | 32.14 |
| Tibacuy | 0.1 | 65 | 17 | 34.93 |
| Tibiritá | 0.1 | 34.7 | 19 | 52.24 |
| Tocaima | 0.4 | 73.3 | 18 | 38.43 |
| Tocancipá | 7.8 | 40.5 | 19 | 50.67 |
| Topaipí | 0.1 | 16.6 | 15 | 34.79 |
| Ubalá | 1.6 | 59.9 | 19 | 40.10 |
| Ubaque | 0.3 | 48.1 | 15 | 26.42 |
| Ubaté | 0.9 | 87.7 | 17 | 44.73 |
| Une | 0.4 | 107.2 | 16 | 30.25 |
| Útica | 0.1 | 67.5 | 20 | 43.02 |
| Venecia | 0.2 | 45.2 | 16 | 38.34 |
| Vergara | 0.2 | 6.6 | 17 | 41.54 |
| Vianí | 0.1 | 25.7 | 17 | 39.71 |
| Villagómez | 0.1 | 9.5 | 10 | 36.43 |
| Villapinzón | 0.8 | 41.8 | 26 | 65.25 |
| Villeta | 0.7 | 56.3 | 16 | 48.21 |
| Viotá | 0.4 | 51.7 | 16 | 33.54 |
| Yacopí | 0.4 | 136.8 | 21 | 35.60 |
| Zipacón | 0.2 | 29.4 | 16 | 35.17 |
| Zipaquirá | 2.8 | 87.9 | 16 | 42.00 |

**Table A3.** Meta department municipalities studied information: MRW, tertiary road network length, and average travel speed.

| Meta Department Municipalities Studied | Municipal Relative Weight (MRW) (%) 2019 | Tertiary Road Network Analyzed (km) | Tertiary Road Network Average Travel Speed (km/h) | Average Travel Speed (km/h) to Department Capital City |
|---|---|---|---|---|
| Villavicencio | 22.9 | 51.8 | 28 | 0.00 |
| Barranca de Upía | 0.6 | 213.8 | 50 | 43.56 |
| Cabuyaro | 3.2 | 330.9 | 29 | 48.46 |
| Castilla la nueva | 8.7 | 253.5 | 20 | 40.52 |
| San Luis de Cubaral | 0.2 | 131.1 | 23 | 42.68 |
| El calvario | 0.1 | 24.7 | 20 | 29.29 |
| El castillo | 0.3 | 365.7 | 30 | 44.52 |
| El dorado | 0.1 | 58.3 | 27 | 42.60 |
| Fuente de oro | 1.4 | 327.8 | 43 | 46.30 |
| La Macarena | 0.6 | 59.35 | 19 | 46.94 |
| La Uribe | 0.2 | 19 | 21 | 37.40 |
| Lejanías | 0.9 | 125.5 | 30 | 50.07 |

**Table A3.** *Cont.*

| Meta Department Municipalities Studied | Municipal Relative Weight (MRW) (%) 2019 | Tertiary Road Network Analyzed (km) | Tertiary Road Network Average Travel Speed (km/h) | Average Travel Speed (km/h) to Department Capital City |
|---|---|---|---|---|
| Mesetas | 0.4 | 167.1 | 29 | 51.48 |
| Puerto Concordia | 0.4 | 15.6 | 38 | 59.54 |
| Puerto Gaitán | 27.8 | 25.5 | 25 | 62.92 |
| Puerto López | 2.8 | 32.6 | 35 | 60.00 |
| Puerto Lleras | 0.7 | 20.6 | 40 | 50.49 |
| Puerto Rico | 0.9 | 75.9 | 46 | 54.82 |
| Restrepo | 0.6 | 28.3 | 34 | 36.92 |
| San Carlos de Guaroa | 1.2 | 55.3 | 34 | 56.81 |
| San Juan de Arama | 0.7 | 28.1 | 35 | 49.93 |
| San Juanito | 0.1 | 9.04 | 19 | 27.19 |
| Vista hermosa | 0.5 | 32.7 | 28 | 53.13 |

**Table A4.** Santander department municipalities studied information: MRW, tertiary road network length, and average travel speed.

| Santander Department Municipalities Studied | Municipal Relative Weight (MRW) (%) 2019 | Tertiary Roadnetwork Analized (km) | Tertiary Roadnetwork Average Travel Speed (km/h) | Average Travel Speed (km/h) to Departament Capital City |
|---|---|---|---|---|
| Bucaramanga | 0.1 | 8.9 | 29 | 0.00 |
| Aguada | 0.1 | 1.9 | 19 | 36.35 |
| Albania | 0.1 | 101.6 | 27 | 40.30 |
| Aratoca | 0.1 | 5.2 | 21 | 37.21 |
| Barbosa | 0.1 | 42.21 | 24 | 42.60 |
| Barichara | 0.1 | 2.55 | 23 | 39.03 |
| Barrancabermeja | 0.1 | 20.7 | 22 | 51.49 |
| Bolívar | 0.1 | 93.1 | 31 | 39.52 |
| Cabrera | 0.1 | 44.8 | 30 | 35.24 |
| Capitanejo | 0.1 | 58.6 | 29 | 31.69 |
| Carcasí | 0.1 | 9.9 | 25 | 29.78 |
| Cerrito | 0.1 | 29.8 | 28 | 32.12 |
| Charalá | 0.1 | 4.7 | 28 | 40.50 |
| Charta | 0.1 | 23.4 | 22 | 29.71 |
| Chima | 0.1 | 31.2 | 22 | 35.18 |
| Chipatá | 0.1 | 67.4 | 27 | 38.89 |
| Cimitarra | 0.1 | 8.6 | 22 | 54.11 |
| Concepción | 0.1 | 18.2 | 27 | 31.73 |
| Confines | 0.1 | 10.9 | 19 | 40.00 |
| Contratación | 0.1 | 42.4 | 27 | 31.40 |
| Coromoro | 0.1 | 74.35 | 23 | 38.32 |
| Curití | 0.1 | 17.5 | 22 | 35.09 |
| El Carmen de Chucuri | 0.1 | 135.9 | 32 | 46.63 |
| El Guacamayo | 0.1 | 37.85 | 32 | 30.60 |
| El Peñón | 0.1 | 67.3 | 23 | 42.63 |
| El Playón | 0.1 | 0.9 | 18 | 36.76 |

**Table A4.** *Cont.*

| Santander Department Municipalities Studied | Municipal Relative Weight (MRW) (%) 2019 | Tertiary Roadnetwork Analized (km) | Tertiary Roadnetwork Average Travel Speed (km/h) | Average Travel Speed (km/h) to Departament Capital City |
|---|---|---|---|---|
| Encino | 0.1 | 33.4 | 21 | 37.79 |
| Enciso | 0.1 | 13.6 | 21 | 31.37 |
| Florián | 0.1 | 6 | 27 | 36.59 |
| Floridablanca | 0.1 | 34.75 | 19 | 33.50 |
| Galán | 0.1 | 13.1 | 20 | 28.49 |
| Gambita | 0.1 | 20.95 | 18 | 41.51 |
| Girón | 0.1 | 47.7 | 25 | 40.00 |
| Guaca | 0.1 | 83.8 | 23 | 29.55 |
| Guadalupe | 0.1 | 26.45 | 24 | 39.92 |
| Guapota | 0.1 | 30.3 | 21 | 40.17 |
| Guavatá | 0.1 | 32.8 | 28 | 42.00 |
| Güepsa | 0.1 | 11.3 | 28 | 42.74 |
| Hato | 0.1 | 95.8 | 18 | 63.48 |
| Jesús María | 0.1 | 78.9 | 27 | 80.67 |
| Jordán | 0.1 | 7.3 | 24 | 33.80 |
| La Belleza | 0.1 | 84.1 | 30 | 37.15 |
| La Paz | 0.1 | 6.6 | 27 | 37.95 |
| Landazuri | 0.2 | 73.1 | 26 | 51.33 |
| Lebrija | 0.2 | 107.2 | 32 | 43.24 |
| Los Santos | 0.2 | 15.3 | 34 | 39.75 |
| Macaravita | 0.2 | 33.6 | 30 | 30.49 |
| Málaga | 0.2 | 6.71 | 43 | 31.12 |
| Matanza | 0.2 | 74.8 | 39 | 30.39 |
| Zapatoca | 0.2 | 29.4 | 26 | 34.58 |
| Mogotes | 0.2 | 37.9 | 26 | 38.79 |
| Molagavita | 0.3 | 37 | 22 | 30.11 |
| Ocamonte | 0.3 | 9.2 | 22 | 40.10 |
| Oiba | 0.3 | 161.3 | 21 | 41.58 |
| Onzaga | 0.3 | 53.9 | 27 | 31.65 |
| Palmar | 0.3 | 50.5 | 28 | 37.25 |
| Villanueva | 0.3 | 78.3 | 34 | 39.23 |
| Palmas del Socorro | 0.3 | 30.1 | 45 | 39.50 |
| Páramo | 0.3 | 135.8 | 30 | 39.44 |
| Piedecuesta | 0.3 | 91.4 | 28 | 46.00 |
| Pinchote | 0.3 | 130.1 | 42 | 39.62 |
| Vélez | 0.3 | 115.8 | 30 | 47.70 |
| Tona | 0.4 | 36.3 | 32 | 27.24 |
| Suratá | 0.4 | 33.2 | 30 | 29.67 |
| Sucre | 0.5 | 563.9 | 30 | 38.73 |
| Suaita | 0.5 | 50.8 | 30 | 42.92 |
| Socorro | 0.8 | 37.4 | 28 | 40.68 |
| Puente Nacional | 0.8 | 99.9 | 34 | 43.22 |
| Puerto Parra | 1 | 66.8 | 33 | 53.26 |
| Simacota | 1 | 27.3 | 31 | 39.23 |

**Table A4.** *Cont.*

| Santander Department Municipalities Studied | Municipal Relative Weight (MRW) (%) 2019 | Tertiary Roadnetwork Analized (km) | Tertiary Roadnetwork Average Travel Speed (km/h) | Average Travel Speed (km/h) to Departament Capital City |
|---|---|---|---|---|
| Puerto Wilches | 1.3 | 13.3 | 32 | 50.77 |
| Santa Bárbara | 1.4 | 46 | 41 | 31.91 |
| San Vicente de Chucurí | 1.4 | 107.5 | 36 | 40.27 |
| San Miguel | 1.5 | 130.9 | 30 | 118.29 |
| Rionegro | 1.7 | 111.2 | 38 | 33.29 |
| Sabana de Torres | 4 | 181.2 | 35 | 54.24 |
| San José de Miranda | 5.2 | 228.5 | 29 | 31.28 |
| San Andrés | 7.2 | 47 | 31 | 30.74 |
| San Benito | 8.3 | 60.2 | 28 | 39.52 |
| San Gil | 26.5 | 324.319048 | 23 | 38.67 |
| San Joaquín | 27.2 | 486.2 | 44 | 33.72 |

**Table A5.** Quindio department municipalities studied information: MRW, tertiary road network length, and average travel speed.

| Quindio Department Municipalities Studied | Municipal Relative Weight (MRW) (%) 2019 | Tertiary Road Network Analyzed (km) | Tertiary Road Network Average Travel Speed (km/h) | Average Travel Speed (km/h) to Department Capital City |
|---|---|---|---|---|
| Armenia | 54 | 5.02 | 25 | 0.00 |
| Buenavista | 1 | 4.02 | 20 | 36.88 |
| Calarcá | 9 | 8.36 | 15 | 22.67 |
| Circasia | 4 | 4.39 | 17 | 33.39 |
| Córdoba | 1 | 3.25 | 15 | 35.60 |
| Filandia | 3 | 3.58 | 22 | 40.00 |
| Génova | 1 | 5.06 | 16 | 35.93 |
| La Tebaida | 8 | 5.85 | 28 | 42.00 |
| Montenegro | 9 | 5.96 | 26 | 35.40 |
| Pijao | 2 | 3.42 | 15 | 33.20 |
| Quimbaya | 7 | 5.98 | 26 | 36.00 |
| Salento | 2 | 6.73 | 24 | 38.05 |

**Table A6.** Colombia departments were analyzed by adding municipalities' information. GDP 2020 in thousands of millions COP.

| Department | Global GDP 2020 | Tertiary Road Network Average Travel Speed (km/h) | Tertiary Road Network Analyzed (km) | Average Travel Speed (km/h) to Department Capital City |
|---|---|---|---|---|
| Casanare | 13.121 | 19 | 4519 | 48.86 |
| Cundinamarca | 61.644 | 18 | 10,526 | 38.77 |
| Meta | 31.363 | 31 | 2452 | 46.63 |
| Santander | 62.570 | 28 | 5510 | 39.02 |
| Quindío | 8.303 | 21 | 62 | 35.88 |

**Table A7.** Colombia's departments' road network lengths by condition in 2020.

| Department | Paved (km) | | | | | Non-Paved (km) | | | | |
|---|---|---|---|---|---|---|---|---|---|---|
| | Very Good | Good | Regular | Poor | Very Poor | Very Good | Good | Regular | Poor | Very Poor |
| Antioquia | 51.37 | 122.24 | 222.38 | 227.11 | 1.00 | 0.10 | 0.25 | 7.73 | 0.10 | 0.10 |
| Atlántico | 18.75 | 4.81 | 2.49 | 0.10 | 0.10 | 0.10 | 3.47 | 8.30 | 28.08 | 2.90 |
| Bolívar | 15.98 | 74.94 | 48.40 | 15.78 | 0.10 | 0.10 | 2.55 | 0.10 | 0.10 | 0.10 |
| Boyacá | 41.44 | 196.93 | 288.15 | 145.44 | 2.00 | 0.10 | 1.21 | 43.64 | 123.85 | 0.09 |
| Caldas | 72.24 | 89.70 | 18.01 | 0.10 | 0.10 | 0.10 | 0.10 | 0.10 | 0.10 | 0.10 |
| Caquetá | 138.70 | 90.23 | 83.06 | 86.31 | 0.10 | 0.10 | 29.24 | 15.05 | 11.87 | 0.10 |
| Casanare | 1.93 | 154.76 | 222.38 | 182.77 | 0.10 | 0.17 | 20.12 | 18.80 | 8.60 | 0.10 |
| Cauca | 105.01 | 176.50 | 243.30 | 118.38 | 1.15 | 2.11 | 97.19 | 265.69 | 259.88 | 9.40 |
| Cesar | 88.08 | 154.82 | 98.80 | 103.86 | 15.98 | 0.10 | 0.10 | 1.00 | 23.90 | 3.49 |
| Choco | 35.72 | 80.78 | 50.99 | 3.78 | 0.10 | 0.10 | 0.10 | 52.43 | 34.03 | 20.45 |
| Córdoba | 51.32 | 33.21 | 68.36 | 90.14 | 0.10 | 2.44 | 13.65 | 13.01 | 16.92 | 0.10 |
| Cundinamarca | 6.41 | 69.54 | 69.95 | 64.03 | 0.10 | 0.10 | 0.58 | 17.37 | 11.27 | 0.10 |
| Guajira | 34.44 | 74.64 | 26.75 | 12.91 | 0.10 | 0.10 | 6.71 | 3.55 | 0.10 | 0.10 |
| Huila | 35.01 | 94.39 | 76.42 | 65.15 | 0.97 | 0.10 | 23.52 | 105.36 | 84.43 | 0.10 |
| Magdalena | 39.58 | 75.89 | 20.32 | 8.19 | 14.47 | 0.10 | 0.10 | 18.02 | 34.66 | 36.82 |
| Meta | 46.02 | 173.31 | 120.62 | 11.85 | 0.88 | 4.05 | 45.63 | 103.44 | 79.08 | 0.10 |
| Nariño | 169.25 | 300.27 | 142.33 | 93.00 | 0.38 | 2.25 | 0.10 | 0.10 | 2.36 | 10.35 |
| Norte de Santander | 19.98 | 140.90 | 135.20 | 131.04 | 0.10 | 0.10 | 0.10 | 72.40 | 34.76 | 6.00 |
| Putumayo | 97.74 | 46.31 | 5.75 | 4.85 | 0.10 | 0.10 | 9.99 | 43.26 | 75.79 | 0.10 |
| Quindío | 39.99 | 30.37 | 40.35 | 2.00 | 0.10 | 0.10 | 0.10 | 0.10 | 0.10 | 0.10 |
| Risaralda | 13.90 | 95.05 | 76.41 | 25.09 | 0.10 | 0.10 | 0.10 | 17.42 | 16.46 | 2.98 |
| Santander | 120.53 | 369.26 | 273.03 | 95.29 | 15.66 | 0.10 | 9.88 | 45.77 | 78.23 | 27.10 |
| Sucre | 35.00 | 24.54 | 32.94 | 45.26 | 14.95 | 0.10 | 0.10 | 0.10 | 0.10 | 0.10 |
| Tolima | 5.85 | 133.06 | 46.56 | 20.54 | 0.10 | 0.10 | 0.10 | 0.10 | 0.10 | 0.10 |
| Valle | 241.10 | 205.87 | 252.64 | 70.22 | 6.66 | 0.10 | 0.10 | 0.10 | 0.10 | 0.10 |
| San Andrés y Providencia | 12.00 | 10.50 | 15.00 | 7.80 | 0.10 | 0.10 | 0.10 | 0.10 | 0.10 | 0.10 |

**Table A8.** Colombia departments' GDP 2020: global and some sectors. Values in thousands of millions COP.

| Department | Global | Manufacturing Industry Sector | Commerce Sector | Agrobusiness Sector | Mining Sector |
|---|---|---|---|---|---|
| Antioquia | 149,666 | 20,586 | 23,856 | 9948 | 5032 |
| Atlántico | 44,923 | 6666 | 8766 | 491 | 104 |
| Bolívar | 34,501 | 5298 | 5045 | 1869 | 890 |
| Boyacá | 27,214 | 3039 | 4416 | 3528 | 1599 |
| Caldas | 17,034 | 1957 | 2757 | 2247 | 421 |
| Caquetá | 4181 | 108 | 687 | 662 | 13 |
| Casanare | 13,121 | 368 | 2514 | 1809 | 4788 |
| Cauca | 18,245 | 2951 | 2038 | 2583 | 190 |
| Cesar | 16,812 | 607 | 2050 | 1814 | 4619 |
| Chocó | 4526 | 31 | 479 | 883 | 930 |
| Córdoba | 18,167 | 1940 | 2336 | 2172 | 751 |
| Cundinamarca | 61,644 | 12,202 | 9281 | 9350 | 411 |
| Guajira | 8093 | 61 | 1236 | 527 | 1484 |

**Table A8.** *Cont.*

| Department | Global | Manufacturing Industry Sector | Commerce Sector | Agrobusiness Sector | Mining Sector |
|---|---|---|---|---|---|
| Huila | 16,810 | 556 | 2695 | 3397 | 805 |
| Magdalena | 13,760 | 518 | 2615 | 2498 | 33 |
| Meta | 31,363 | 770 | 3739 | 4413 | 13,089 |
| Nariño | 15,838 | 358 | 2952 | 2749 | 135 |
| Norte de Santander | 15,798 | 804 | 2741 | 1811 | 374 |
| Putumayo | 3331 | 29 | 564 | 247 | 646 |
| Quindío | 8303 | 360 | 1643 | 1603 | 19 |
| Risaralda | 16,605 | 2030 | 3283 | 1335 | 82 |
| Santander | 62,570 | 10,208 | 8410 | 6985 | 1886 |
| Sucre | 8444 | 537 | 1329 | 843 | 50 |
| Tolima | 21,621 | 1896 | 3459 | 3905 | 598 |
| Valle | 100,169 | 15,614 | 16,588 | 7142 | 126 |
| San Andrés y Providencia | 1312 | 17 | 688 | 25 | 1 |

**Table A9.** Colombia department GDP 2020, and primary and tertiary roadway network length.

| Department | Global GDP Thousands of Million COP | Tertiary Roadway Network (km) | Primary Roadway Network (km) |
|---|---|---|---|
| Antioquia | 149,666 | 12,500 | 1.539 |
| Atlántico | 44,923 | 1250 | 340 |
| Bolívar | 34,501 | 6250 | 548 |
| Boyacá | 27,214 | 14,000 | 1.054 |
| Caldas | 17,034 | 2500 | 378 |
| Caquetá | 4181 | 4900 | 440 |
| Casanare | 13,121 | 3750 | 768 |
| Cauca | 18,245 | 6875 | 1.429 |
| Cesar | 16,812 | 1875 | 723 |
| Choco | 4526 | 625 | 270 |
| Córdoba | 18,167 | 6875 | 749 |
| Cundinamarca | 61,644 | 13,125 | 1.103 |
| Guajira | 8093 | 2250 | 357 |
| Huila | 16,810 | 8125 | 838 |
| Magdalena | 13,760 | 4375 | 549 |
| Meta | 31,363 | 5000 | 1.019 |
| Nariño | 15,838 | 5000 | 793 |
| Norte de Santander | 15,798 | 3700 | 667 |
| Putumayo | 3331 | 1200 | 399 |
| Quindío | 8303 | 1875 | 138 |
| Risaralda | 16,605 | 1900 | 300 |
| Santander | 62,570 | 7600 | 966 |
| Sucre | 8444 | 3000 | 371 |
| Tolima | 21,621 | 9375 | 877 |
| Valle | 100,169 | 4375 | 862 |
| San Andrés y Providencia | 1312 | 0.05 | 50 |

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
