# Peer review of "Correlation Analysis between Roadway Networks and Economic Ranking—Case Study: Municipalities and Departments of Colombia"

_infrastructures, doi:10.3390/infrastructures7090118_

Round 1

Reviewer 1 Report (Previous Reviewer 1)

The authors have revised the manuscript. I agree the manuscript to be published. 

Author Response

Point 1: The authors have revised the manuscript. I agree the manuscript to be published

Response: We appreciate and thank you for your valuable contribution to our development as researchers and specifically to the improvement of the document.

We are sending this message to inform you that we have completed the reviewer 3 of the article, and we have uploaded the latest version on the platform with the point-by-point responses as presented below:

Point 1: The paper is fraught with spelling mistakes and I intentionally skip pointing each of these out as it is not worth the time. The authors are strongly advised to proofread the manuscript before resubmitting it to this journal or elsewhere

 Response 1: The new version of the manuscript has been revised and significantly modified in terms of spelling and grammar.

Point 2: The authors need to focus on the references. For example, reference number 4 (Jalan and Ravallion) is published in 2002 (Vol 17, Iss 4) but the authors reference it as Forthcoming. Also, reference number 13 must be Ghosh and Dinda (Ln 109)

Response 2: The bibliographic references were reviewed regarding the publication date and the wording of the respective text in past conjugation.

Point 3: The most serious flaw of the paper in my opinion is it being statistically sound. For a parametric study such as this (assuming the authors relied on normal distributions), a reader would expect to see a word mentioned about the data compliant with normality (e.g. Shapiro-Wilk test) or residual analysis (with residuals being randomly scattered and not following any pattern). My main concern is with multicollinearity as the authors indicate for a few tests the independent variables to be speed, time, and distance. There is a direct relationship between these and hence can render an analysis incorrect if not checked for multicollinearity. Moreover, the authors are advised to rely on advanced statistical tools such as R or SPSS compared to MS Excel. Lastly, I advise the authors to also consider using R instead of R2 throughout the paper as the former is a better indicator of the correlation including its direction (positive or negative). R2 is definitely a better source to indicate the magnitude of variance but R of the direction and magnitude of the correlation. The authors must also state clearly in the body of text that correlation does not necessarily imply causation

Response 3: The analysis of the correlation coefficient R has been added to the study in departmental data, validating the statistical behavior of the correlations presented based on the value of R2. Tables 7 and 8 have been added.

Calculations of R and R2 were made using the "R" software and compared with those obtained in Microsoft Excel. The data presented in the manuscript are validated since they were the same.

The departmental and municipal data were tested about normal distribution, using Shapiro Wilk and d’Agostino Pearson test, using the Real Statistics application for Microsoft Excel.

In the discussion section, it has been clarified to the reader that if there is a correlation between 2 variables, it doesn´t imply that the independent variable explain the behavior of the dependent variables; that is, there is not necessarily a causal relationship between the 2 variables. Therefore, the explanations of the relationships between variables, exposed in this discussion, stem from social and economic principles and the state of the art and don’t pretend to be supported only by validated statistical correlations.

About the Multicollinearity, the condition has been manifested in the manuscript in the second paragraph of page 9. There it´s commented that when noticing the multicollinearity in departmental analysis the variable speed was ruled out. In the analysis section by municipalities, the variable speed did not present multicollinearity because was data registered in the field, and no was the result of calculus between distance and time data

Point 4: The Introduction section must introduce the reader to the problem first by explaining why it makes sense to address the issue in a broader context. Then drill down further explaining how the current work helps address this problem in the broader context and how this research can be beneficial in the future. Following the Introduction section, a Literature Review section is highly recommended. The Introduction section of the current manuscript very closely resembles a Literature Review section in my opinion. Also, a hypothesis section is usually included at the end of the introduction section and not the methods section, although it is not a requirement.

Response 4: The manuscript introduction was enlarged as suggested.

Again, we appreciate the input of all reviewers.

Sincerely, 

Reviewer 2 Report (Previous Reviewer 2)

The manuscript has been modified according to the comments. And the creative points have been concluded carefully. Moreover, the conclusions have been summarized in the article. So I think this article can be accepted in present form.

Author Response

First of all, we appreciate and thank you for your valuable contribution to our development as researchers and specifically to the improvement of the document.

Regarding the changes made according to the reviewer 3 suggested contributions, we mention that:

Point 1: The paper is fraught with spelling mistakes and I intentionally skip pointing each of these out as it is not worth the time. The authors are strongly advised to proofread the manuscript before resubmitting it to this journal or elsewhere

Response 1: The new version of the manuscript has been revised and significantly modified in terms of spelling and grammar.

Point 2: The authors need to focus on the references. For example, reference number 4 (Jalan and Ravallion) is published in 2002 (Vol 17, Iss 4) but the authors reference it as Forthcoming. Also, reference number 13 must be Ghosh and Dinda (Ln 109)

Response 2: The bibliographic references were reviewed regarding the publication date and the wording of the respective text in past conjugation.

Point 3: The most serious flaw of the paper in my opinion is it being statistically sound. For a parametric study such as this (assuming the authors relied on normal distributions), a reader would expect to see a word mentioned about the data compliant with normality (e.g. Shapiro-Wilk test) or residual analysis (with residuals being randomly scattered and not following any pattern). My main concern is with multicollinearity as the authors indicate for a few tests the independent variables to be speed, time, and distance. There is a direct relationship between these and hence can render an analysis incorrect if not checked for multicollinearity. Moreover, the authors are advised to rely on advanced statistical tools such as R or SPSS compared to MS Excel. Lastly, I advise the authors to also consider using R instead of R2 throughout the paper as the former is a better indicator of the correlation including its direction (positive or negative). R2 is definitely a better source to indicate the magnitude of variance but R of the direction and magnitude of the correlation. The authors must also state clearly in the body of text that correlation does not necessarily imply causation

Response 3: The analysis of the correlation coefficient R has been added to the study in departmental data, validating the statistical behavior of the correlations presented based on the value of R2. Tables 7 and 8 have been added.

Calculations of R and R2 were made using the "R" software and compared with those obtained in Microsoft Excel. The data presented in the manuscript are validated since they were the same.

The departmental and municipal data were tested about normal distribution, using Shapiro Wilk and d’Agostino Pearson test, using the Real Statistics application for Microsoft Excel.

In the discussion section, it has been clarified to the reader that if there is a correlation between 2 variables, it doesn´t imply that the independent variable explain the behavior of the dependent variables; that is, there is not necessarily a causal relationship between the 2 variables. Therefore, the explanations of the relationships between variables, exposed in this discussion, stem from social and economic principles and the state of the art and don’t pretend to be supported only by validated statistical correlations.

About the Multicollinearity, the condition has been manifested in the manuscript in the second paragraph of page 9. There it´s commented that when noticing the multicollinearity in departmental analysis the variable speed was ruled out. In the analysis section by municipalities, the variable speed did not present multicollinearity because was data registered in the field, and no was the result of calculus between distance and time data

Point 4: The Introduction section must introduce the reader to the problem first by explaining why it makes sense to address the issue in a broader context. Then drill down further explaining how the current work helps address this problem in the broader context and how this research can be beneficial in the future. Following the Introduction section, a Literature Review section is highly recommended. The Introduction section of the current manuscript very closely resembles a Literature Review section in my opinion. Also, a hypothesis section is usually included at the end of the introduction section and not the methods section, although it is not a requirement.

Response 4: The manuscript introduction was enlarged as suggested.

Please see the attachment of the final version of the paper with all the changes mentioned here.

Again, we appreciate the input of all reviewers.

Sincerely, 

Reviewer 3 Report (New Reviewer)

The authors present an interesting study on the statistical correlation of GDP and Municipal Relative Weights with various variables related to the economy and road conditions. The authors did a good job of presenting previous work done in the area and how the current study contributes to the body of work. However, I am compelled to reject the paper in its current form due to the following reasons:

1) The paper is fraught with spelling mistakes and I intentionally skip pointing each of these out as it is not worth the time. The authors are strongly advised to proofread the manuscript before resubmitting it to this journal or elsewhere.

2) The authors need to focus on the references. For example, reference number 4 (Jalan and Ravallion) is published in 2002 (Vol 17, Iss 4) but the authors reference it as Forthcoming. Also, reference number 13 must be Ghosh and Dinda (Ln 109).

3) The most serious flaw of the paper in my opinion is it being statistically sound. For a parametric study such as this (assuming the authors relied on normal distributions), a reader would expect to see a word mentioned about the data compliant with normality (e.g. Shapiro-Wilk test) or residual analysis (with residuals being randomly scattered and not following any pattern). My main concern is with multicollinearity as the authors indicate for a few tests the independent variables to be speed, time, and distance. There is a direct relationship between these and hence can render an analysis incorrect if not checked for multicollinearity. Moreover, the authors are advised to rely on advanced statistical tools such as R or SPSS compared to MS Excel. Lastly, I advise the authors to also consider using R instead of R2 throughout the paper as the former is a better indicator of the correlation including its direction (positive or negative). R2 is definitely a better source to indicate the magnitude of variance but R of the direction and magnitude of the correlation. The authors must also state clearly in the body of text that correlation does not necessarily imply causation.

4) The Introduction section must introduce the reader to the problem first by explaining why it makes sense to address the issue in a broader context. Then drill down further explaining how the current work helps address this problem in the broader context and how this research can be beneficial in the future. Following the Introduction section, a Literature Review section is highly recommended. The Introduction section of the current manuscript very closely resembles a Literature Review section in my opinion. Also, a hypothesis section is usually included at the end of the introduction section and not the methods section, although it is not a requirement.

I hope the authors find the above suggestions useful to improve the manuscript thereby yielding an important contribution to the body of work.

Author Response

Respected reviewer.

First of all, we appreciate and thank you for your valuable contribution to our development as researchers and specifically to the improvement of the document.

Regarding the changes made according to your suggested contributions, we mention that:

The new version of the manuscript has been revised and significantly modified in terms of spelling and grammar, sorry for all mistakes in the previous version .

The analysis of the correlation coefficient R has been added to the study in departmental data, validating the statistical behavior of the correlations presented based on the value of R2. Tables 7 and 8 have been added.

Calculations of R and R2 were made using the "R" software and compared with those obtained in Microsoft Excel. The data presented in the manuscript are validated since they were the same.

The bibliographic references were reviewed regarding the publication date and the wording of the respective text in past conjugation.

The departmental and municipal data were tested about normal distribution, using Shapiro Wilk and d’Agostino Pearson test, using the Real Statistics application for Microsoft Excel.

In the discussion section, it has been clarified to the reader that if there is a correlation between two variables, it does not imply that the independent variable explain the behavior of the dependent variables; that is, there is not necessarily a causal relationship between the two variables. Therefore, the explanations of the relationships between variables, exposed in this discussion, stem from social and economic principles and the state of the art and do not pretend to be supported only by validated statistical correlations. 

About the Multicollinearity, the condition has been manifested in the manuscript in the second paragraph of page 9. There it´s commented that when noticing the multicollinearity in departmental analysis the variable speed was ruled out. In the analysis section by municipalities, the variable speed did not present multicollinearity because was data registered in the field, and no was the result of calculus between distance and time data.

The manuscript introduction was enlarged as you suggested.

Again, we appreciate the input of all reviewers.

Sincerely,

Round 2

Reviewer 3 Report (New Reviewer)

The authors have significantly improved the quality of the manuscript and the reviewer feels the revised manuscript is ready for publication.

This manuscript is a resubmission of an earlier submission. The following is a list of the peer review reports and author responses from that submission.

Round 1

Reviewer 1 Report

The analysis provided in the manuscript has not led to robust and concrete results. All the mentioned parameters have no shown any correlations between them. Tables with the original data are not provided in the paper. 

Reviewer 2 Report

Because there are no frequent studies that analyze a correlation between the quantity and condition of roadway networks of a municipality or department with variables that measure the economic level within a territory. Aiming at this point, this paper contributes to the state-of-the-art, by attempting to correlate travel conditions (time, average speed) and roadway length, with economic development indicators such as departmental Gross Domestic Product and Municipal Relative for municipalities. The most important is that this point is not creative, so the authors should conclude it carefully.

Besides, the materials have introduced the problems, but the data has not been presented in this research. The data analysis has been conducted in this research and the analysis results also have been presented in pictures. But the data sample should be introduced in the manuscript.

In addition, the methodology is so simple that the advantages of this research method can’t show up. Moreover, the conclusions should be summarized in the article, which can’t be added to the discussions.